# Disentangling microbial networks across pelagic zones in the tropical and subtropical global ocean

Ina M. Deutschmann [1] ✉, Erwan Delage [2,3], Caterina R. Giner [1], Marta Sebastián [1], Julie Poulain[4], Javier Arístegui [5], Carlos M. Duarte [6], Silvia G. Acinas [1], Ramon Massana [1], Josep M. Gasol [1], Damien Eveillard [2,3], Samuel Chaffron [2,3] & Ramiro Logares [1] ✉

Microbial interactions are vital in maintaining ocean ecosystem function, yet their dynamic nature and complexity remain largely unexplored. Here, we use association networks to investigate possible ecological interactions in the marine microbiome among archaea, bacteria, and picoeukaryotes throughout different depths and geographical regions of the tropical and subtropical global ocean. Our findings reveal that potential microbial interactions change with depth and geographical scale, exhibiting highly heterogeneous distributions. A few potential interactions were global, meaning they occurred across regions at the same depth, while 11-36% were regional within specific depths. The bathypelagic zone had the lowest proportion of global associations, and regional associations increased with depth. Moreover, we observed that most surface water associations do not persist in deeper ocean layers despite microbial vertical dispersal. Our work contributes to a deeper understanding of the tropical and subtropical global ocean interactome, which is essential for addressing the challenges posed by global change.

Microorganisms play fundamental roles in ocean ecosystem functioning and global biogeochemical cycles[1–3]. Changes in the composition of the ocean microbiome can affect ecosystem function. Therefore, we must understand the mechanisms driving community change. The main processes shaping microbial community composition are selection, dispersal, and drift[4]. Selection exerted via abiotic environmental heterogeneity and biotic interactions is essential in structuring the ocean microbiome, leading to heterogeneities in community composition that can reflect those found in the ocean, generally related to temperature, light, pressure, nutrients, and salinity. Global-scale studies of the surface ocean reported strong associations between microbial community composition and temperature[5–9]. In addition, we previously found that temperature-driven selection seems to be a major factor influencing co-occurrence networks in surface ocean prokaryotes[5]. Marked changes in microbial communities with depth have also been reported[10–15], reflecting the steep vertical gradients in light, temperature, nutrients, and pressure in the ocean.

The ocean microbiome contains prokaryotes (bacteria and archaea) and unicellular eukaryotes, which fundamentally differ in ecological roles, functional versatility, and evolutionary history[16] and are connected through biogeochemical, and food web interaction networks[17,18]. Still, our knowledge about their ecological interactions remains limited, even though these interactions sustain marine food

[1]Institute of Marine Sciences (ICM), CSIC, Barcelona, Spain. [2]Nantes Université, CNRS UMR 6004, LS2N, F-44000 Nantes, France. [3]Research Federation for the study of Global Ocean Systems Ecology and Evolution, FR2022/Tara Oceans GOSEE, Paris, France. [4]Génomique Métabolique, Genoscope, Institut François Jacob, CEA, CNRS, Univ Evry, Université Paris-Saclay, Evry, France. [5]Instituto de Oceanografía y Cambio Global, IOCAG, Universidad de Las Palmas de Gran Canaria, ULPGC, Gran Canaria, Spain. [6]King Abdullah University of Science and Technology (KAUST), Red Sea Research Center (RSRC), Thuwal, Saudi Arabia. ✉e-mail: ina.m.deutschmann@gmail.com; ramiro.logares@icm.csic.es

webs and contribute to nutrient recycling in the oceans[3,19]. Microbial interactions are challenging to be resolved experimentally, mainly because most microorganisms are hard to cultivate[20,21], and synthetic laboratory communities are unlikely to mirror the complexity of wild communities. However, association networks inferred from omics data can potentially unravel microbial interactions.

Microbial association networks based on abundance data represent putative ecological interactions that laboratory experiments must confirm. Nevertheless, association networks are one of the best tools to address the large complexity of microbial interactions. Association networks can provide a general overview of the potential microbial interactions in the ocean aggregated over a given period[10,11,22–26] or through space[27–29]. Here, we refer to this set of potential ecological interactions based on association networks as the "interactome", similarly to other works[27,29]. Previous work investigated marine microbial associations within and across depths. For example, prokaryotic associations were investigated in the San Pedro Channel, off the coast of Los Angeles, California, covering the water column from the surface (5 m) to the seafloor (890 m)[10,11]. Furthermore, a global survey from the TARA Oceans expedition investigated planktonic associations between a range of organismal size fractions in the epipelagic zone[27,29]. However, these studies did not include the bathypelagic realm below 1000 m depth, representing one of the largest microbial habitats on the planet[30].

Microbial interaction networks change over space[29] and time[31], yet our knowledge about their dynamics is poor, limiting our capacity to comprehend how robust or fragile the interactions that sustain ocean food webs are. Previous studies have investigated microbial associations in the ocean over space using static networks determined from spatially distributed samples, which capture global, regional, and local associations in a single network[27,28,32]. Furthermore, static networks derived from global ocean expeditions include temporal associations since samples are collected over several months. Disentangling them from spatial associations is challenging and would need comprehensive spatiotemporal sampling campaigns. Nevertheless, determining how networks change across the ocean can help us understand the biogeography of interactions by showing which ones are cosmopolitan and which are restricted to specific regions or depths.

Spatially widespread or global associations may be part of the core microbiome, defined as the set of interacting microbes essential for the functioning of the ocean ecosystem[33]. Core associations may be detected by constructing a single network from numerous locations and identifying the most significant and strongest associations[34]. Once determined, core microbial associations could be the target of future monitoring and conservation efforts. On the other hand, regional or local associations may reflect interactions occurring in specific locations due to taxa distributions resulting from abiotic or biotic environmental selection or dispersal limitation. Regional networks could also contribute to determining stable (i.e., two partners always together) or variable (one partner able to interact with multiple partners across locations) associations. The fraction of regional associations may be determined by excluding all samples belonging to one region, recomputing network inference with the reduced dataset, and examining which associations are missing[27]. Alternatively, regional networks are computed considering samples belonging to the regions, allowing to determine global and regional associations[35] by investigating which edges are common and which are unique. However, this approach for determining regional networks requires many samples per delineated zone, which may not be available due to logistic or budgetary limitations during sampling campaigns. A recent approach we developed circumvents this limitation by deriving sample-specific subnetworks from a single static, i.e., all-sample network, which allows quantifying association recurrence over spatiotemporal scales[36]. In a nutshell, each subnetwork (sample) includes only nodes and edges present in the overarching static network. Three key conditions must be met for an

edge to be included in a subnetwork: 1) the edge must already exist in the single static network, 2) both microorganisms connected by the edge must have a sequence abundance above zero in the sample, and 3) the microorganisms must appear together in more than 20% of the samples for a specific marine region and depth. Here, we apply this innovative approach to investigate the biogeography of microbial interactions in the tropical and subtropical global ocean and the Mediterranean Sea, considering horizontal and vertical dimensions (including the deep sea).

In this work, we ask: Are regional associations more prevalent than global or cosmopolitan associations? Are there associations only found in specific regions or depths? What are the main changes in associations across the water column? Are the distributions of taxa and that of associations coupled? To address these questions, we analyze associations between archaea, bacteria, and picoeukaryotes using a unique dataset including 397 samples covering the water column, from surface to deep waters (up to ~4000 m depth), in the Mediterranean Sea (hereafter MS) and tropical and subtropical areas of five ocean basins: North and South Atlantic Ocean, North and South Pacific Ocean, and Indian Ocean (hereafter NAO, SAO, NPO, SPO, and IO respectively) (Fig. 1). Our exploration of association networks across regions and depths allows us to determine global and regional associations, starting to unveil the potential biogeography of the ocean interactome.

## Results

### Network architecture changed along the water column

Microbial dispersal, as well as vertical and horizontal environmental heterogeneity, are expected to affect microbial communities and network topologies. We found vertical and horizontal differences in the distributions of Amplicon Sequence Variants (ASVs) and the number of unique ASVs per depth layer (Supplementary Table 1, Supplementary Fig. 1; see specific details in the Zenodo repository[37] [section 06_VerticalConnectivity, Additional Tables]), which is consistent with previous works[5,12,38–40]. Contrary to communities, we have a limited understanding of how much marine microbial networks change due to dispersal as well as vertical and horizontal environmental heterogeneity. Analyzing the topology of subnetworks from specific ocean regions and depths is the first step to addressing this issue. We generated 397 sample-specific subnetworks and compared them across regions and depth layers using eight network metrics (see Methods). We found that network metrics changed along the water column, with surface networks tending to display higher values, and a higher variability, in the number of nodes and edges in the ocean basins (Supplementary Fig. 2). As a general trend, subnetworks from deeper zones were more clustered (higher transitivity), had higher average path length, featured stronger associations (average positive association scores), and had lower assortativity (based on the degree) compared to those in surface waters. In addition, most subnetworks from the Deep Chlorophyll Maximum (DCM) and bathypelagic zones had the highest edge density, i.e., the highest node connectivity. In contrast, in the MS, the surface subnetworks had the highest node connectivity (Supplementary Fig. 2).

### Only a few global associations

We computed the spatial recurrence, i.e., prevalence, of each association as the fraction of subnetworks in which a given association was present across all 397 subnetworks (Fig. 2a) and within each region-depth-layer combination (Fig. 2b). The tropical and subtropical global ocean surface layer (contributing 40% of the samples) had more associations than the other depths (Fig. 2b). Remarkably, 14971 of 18234 (82.1%) surface ocean associations detected in the basins were absent in the MS. In turn, the number of surface associations was similar across the five ocean basins (Fig. 2b).

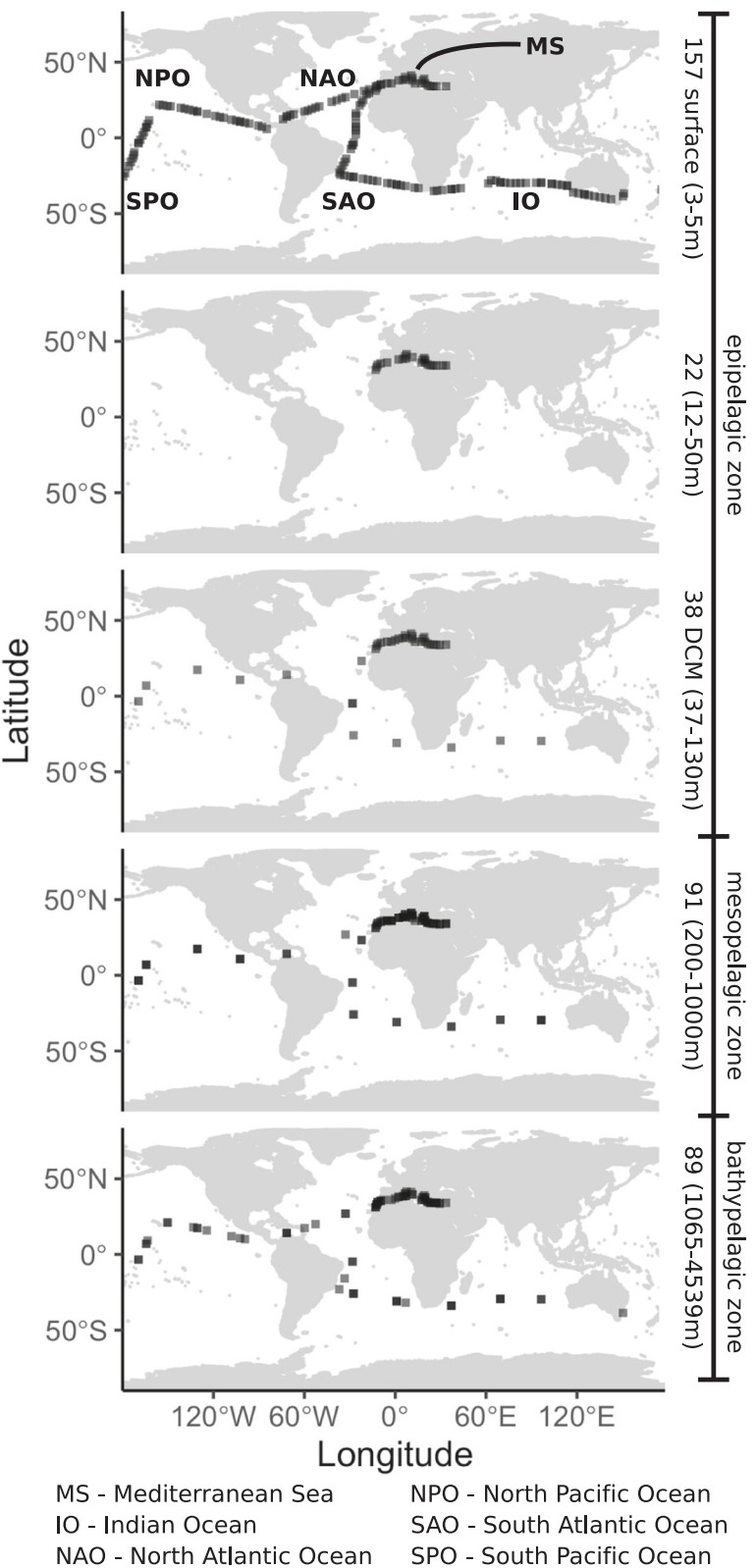

**Fig. 1 | Sampling scheme.** Location, number, and depth range of samples from the epipelagic zone, including surface and DCM layers, the mesopelagic zone, and the bathypelagic zone from the global tropical and subtropical ocean and the Mediterranean Sea. Source data are provided in the GitHub/Zenodo[37] repositories (sections 00_Tables and 01_Metadata; see Data Availability). The map was obtained from the package 'ggplot2'[102].

Highly prevalent associations present across all regions and linked to high or low abundance ASVs are candidates to represent putative core interactions in the tropical and subtropical global ocean that may be connected to processes that are important for ecosystem function. We defined global associations as those appearing in more than 70% of the subnetworks in each region. In addition, we resolved prevalent (≤70% and >50%) and low-frequency (≤50% and >20%) associations. The MS is a distinct region compared

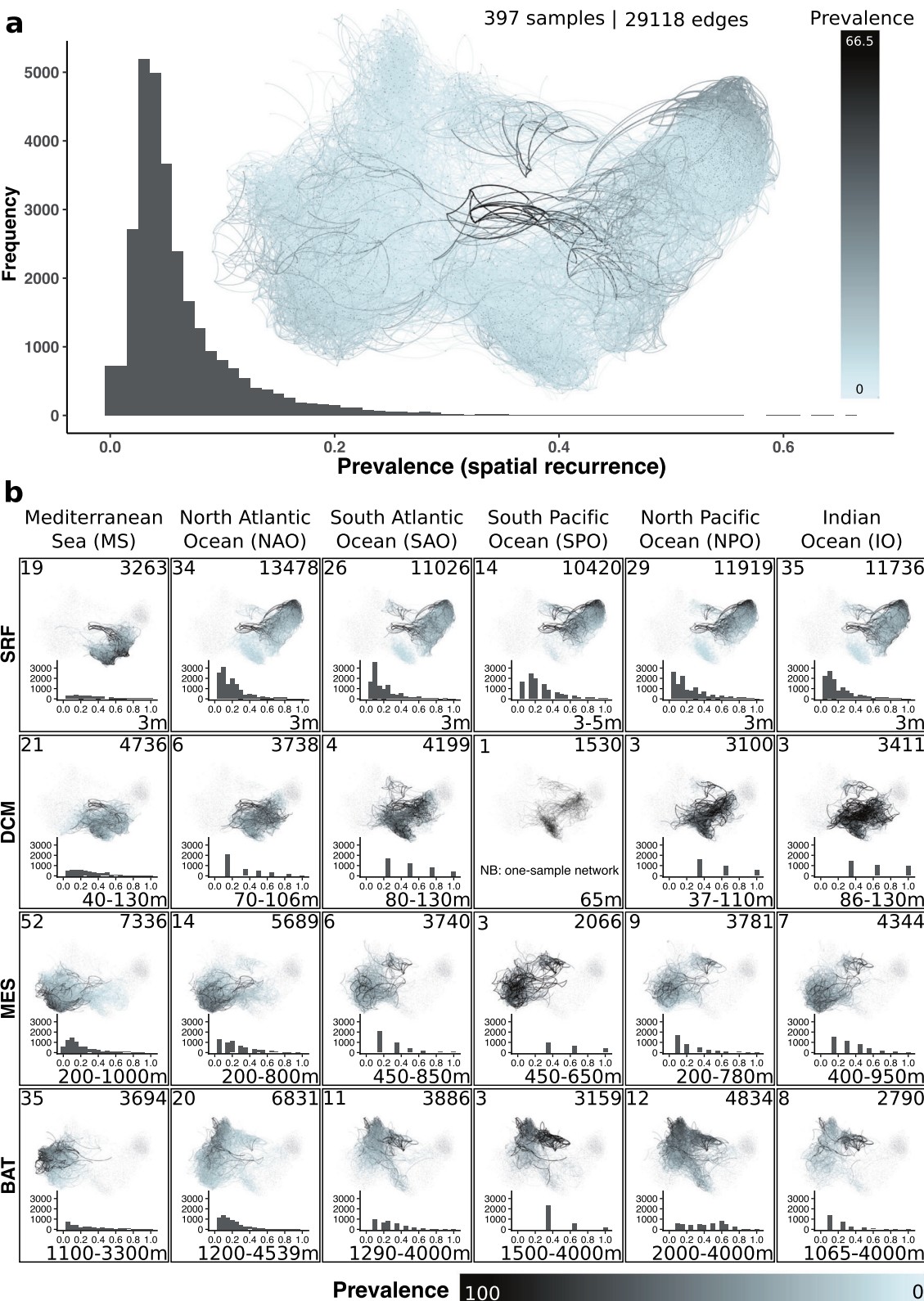

**Fig. 2 | Spatial recurrence of potential microbial interactions. a** Association prevalence, that is, the fraction of subnetworks (samples) in which an association appeared considering all depth layers across the tropical and subtropical global ocean and the Mediterranean Sea. Associations that occurred more often (black) appeared in the middle of the single static network visualization. Most edges had a low prevalence (blue) <20%. **b** The sample-specific subnetworks of the four depth layers (rows): surface (SRF), DCM, mesopelagic (MES), and bathypelagic (BAT), in the five oceanic basins and the Mediterranean Sea (columns). The histograms show the association prevalence within each depth layer and region (excluding absent associations, i.e., 0% prevalence). The number of samples appears in the upper left corner, the number of edges with a prevalence >0% in the upper right corner, and the depth range in the lower right corner (in m below surface). Note that the prevalence rises to 100% in **b** vs. 66.5% in **a**. Source data are provided in the GitHub/Zenodo[37] repositories (sections 02_NetworkConstruction and 04_Prevalence; see Data Availability).

**Table 1 | Classified associations per depth layer**

| Association classification | Epipelagic (Surface) | Epipelagic (DCM) | Mesopelagic | Bathypelagic |
|---|---|---|---|---|
| Global | 26 (0.14%) | 23 (0.31%) | 21 (0.20%) | - |
| Prevalent | 22 (0.12%) | 47 (0.64%) | 10 (0.10%) | 7 (0.07%) |
| Low-frequency | 105 (0.58%) | 160 (2.17%) | 212 (2.05%) | 51 (0.51%) |
| Global (no MS) | 86 (0.47%) | 52 (0.70%) | 28 (0.27%) | 9 (0.09%) |
| Prevalent (no MS) | 207 (1.14%) | 76 (1.03%) | 27 (0.26%) | 28 (0.28%) |
| Low-frequency (no MS) | 1361 (7.46%) | 219 (2.97%) | 342 (3.30%) | 489 (4.84%) |
| Regional | 2014 (11.05%) | 2290 (31.03%) | 3420 (33.00%) | 3669 (36.33%) |
| MS | 596 (3.27%) | 1295 (17.55%) | 2254 (21.75%) | 1217 (12.05%) |
| NAO | 577 (3.16%) | 306 (4.15%) | 422 (4.07%) | 1522 (15.07%) |
| SAO | 162 (0.89%) | 304 (4.12%) | 301 (2.90%) | 143 (1.42%) |
| SPO | 152 (0.83%) | 105 (1.42%) | 40 (0.39%) | 109 (1.08%) |
| NPO | 298 (1.63%) | 133 (1.80%) | 204 (1.97%) | 516 (5.11%) |
| IO | 229 (1.26%) | 147 (1.99%) | 199 (1.92%) | 162 (1.60%) |
| Other[a] | 16067 (88.12%) | 4860 (65.85%) | 6701 (64.66%) | 6372 (63.10%) |
| Other (no MS)[a] | 14566 (79.88%) | 4743 (64.27%) | 6547 (62.17%) | 55904 (58.46%) |
| Present | 18234 (100%) | 7380 (100%) | 10364 (100%) | 10099 (100%) |
| Absent | 10884 | 21738 | 18754 | 19019 |

The sum of classified associations (including Other) is the number of present associations. Absent associations appear in other layers but in no subnetwork of a given layer. Global, prevalent, and low-frequency associations have been computed with and without considering the MS. The proportion of regional associations increased with depth.

[a]The number of unclassified (Other) associations is computed from present, regional, global, prevalent, and low-frequency associations. The last three classifications have been done with and without the MS, and subsequently, the number of unclassified (other) associations varies.

with the ocean basins. For instance, the bathypelagic is warmer (median temperature of 13.8 °C) than the ocean basins' bathypelagic zone (median temperature between 1.4 °C in SPO and 4.4 °C in NAO). Thus, we characterized associations for all six regions and the ocean basins only. We found slightly to moderately more global, prevalent, and low-frequency associations when not considering the MS (Table 1, Supplementary Fig. 3). The fraction of global, prevalent, and low-frequency associations was highest in the DCM layer and lowest in the bathypelagic zone (Table 1). Specifically, while we found several (28–86 no MS, and 21–26 with MS) global associations in the epi- and mesopelagic zones, only a few or none (9 no MS, and none with MS) global associations were identified in the bathypelagic zone. While Alphaproteobacteria were present in associations across depth layers, they dominated in the epipelagic global associations (Supplementary Fig. 3). Still, Alphaproteobacteria were well represented in global associations in the bathypelagic (Supplementary Fig. 3). Dinoflagellates were also well represented in epipelagic global associations. Most global associations from the mesopelagic and bathypelagic included Thaumarchaeota, which were more abundant in deeper zones (Supplementary Figure 3). Dinoflagellates and Alphaproteobacteria tended to be common among epipelagic Prevalent and Low-Frequency associations, while other taxonomic groups displayed a more variable representation (Supplementary Fig. 3). In the mesopelagic and bathypelagic, Thaumarchaeota and Alphaproteobacteria tended to be common among Prevalent and Low-Frequency associations, yet other lineages were prevalent in specific cases, such as the eukaryotic SAR, Dinoflagellates, Radiolarians, Actinobacteria, Deltaproteobacteria, and Gammaproteobacteria (See more details in Supplementary Fig. 3 and in the Zenodo repository[37] [05_ClassifyingAssociations; here all edges are listed via their ASVs and their classifications]).

## Consistent high-rank taxonomy of associations across regions

Next, we considered the most prevalent associations within a specific region and depth, i.e., those found in over 70% of the subnetworks of one region and depth layer. Despite the few global associations determined before, we found that high-rank taxonomic patterns of associated taxa were consistent across the water column in different regions (Fig. 3). The epipelagic layers (surface and DCM) and the two lower layers (meso- and bathypelagic zones) were more similar to each other, respectively (Fig. 3). The fraction of associations including Alphaproteobacteria was moderate to high in all zones in contrast to Cyanobacteria appearing mainly, as expected, in the epipelagic zone (Fig. 3, Supplementary Data 1). The fraction of associations including Dinoflagellata was moderate to high in the epipelagic zone and lower in the meso- and bathypelagic zones (Fig. 3, Supplementary Data 1). While Dinoflagellata associations dominated most epipelagic layers, fewer were found in the MS and SAO surface waters and the NAO DCM (Fig. 3, Supplementary Data 1). Thaumarchaeota associations were moderate to high in the mesopelagic (dominant in the MS), moderate in the bathypelagic, and low in the epipelagic zone (Fig. 3, Supplementary Data 1). Associations including Gammaproteobacteria increased with depth, being higher in the meso- and bathypelagic than in the epipelagic, especially in the SAO, SPO, NPO, and IO (Fig. 3, Supplementary Data 1). Above, we have only described main patterns; see further details in the Zenodo repository[37], sections 04_Prevalence, 05_ClassifyingAssociations, and 06_VerticalConnectivity).

## The proportion of regional associations increased with depth

We determined regional associations within each depth layer. Regional associations were defined as those detected in at least one sample-specific subnetwork from one region, being absent from all subnetworks of the other five regions. Results indicated an increasing proportion of regional associations with depth (Table 1, Fig. 4a, b, Supplementary Fig. 4). We found substantially more associations in the DCM and mesopelagic layers of the MS than in corresponding layers of the ocean basins. The previous may reflect the different characteristics of these layers in the MS vs. the ocean basins or the massive differences in spatial dimensions between the ocean basins and the MS. More surface and bathypelagic regional associations were found in the MS and NAO than in other regions (Table 1). Most regional associations had low prevalence, i.e., they were present in a few sample-specific subnetworks within the region (Fig. 4c). We found 235 highly prevalent (>70%) regional associations among prokaryotes, 89 among eukaryotes, and 24 between domains (Supplementary Data 2).

## Few associations were present throughout the water column

Previous studies have found substantial vertical connectivity in the ocean microbiota, with surface microorganisms impacting deep-sea counterparts[12,41]. Thus, we analyzed the vertical connectivity of potential microbial interactions to determine what surface associations could be detected along the water column. Few associations were present throughout the water column within a region, including 327 among prokaryotes, 119 among eukaryotes, and 13 between domains (Supplementary Data 3). In general, most associations from the meso- and bathypelagic did not appear in the upper layers except for the MS and NAO, where most and about half, respectively, of the bathypelagic associations already appeared in the mesopelagic (Fig. 5). Specifically, 81.8–90.9% of the mesopelagic and 43.5–72.7% of the bathypelagic associations appeared for the first time in these layers when the five ocean basins were considered (Supplementary Table 2). In the MS, 71.2% of the mesopelagic and 22.4% of the bathypelagic associations appeared for the first time in these layers. We found that 69.7% of the associations in the bathypelagic zone already appeared in the mesopelagic zone (Supplementary Table 2). This points to specific microbial interactions in the deep ocean that do not occur in the upper layers. In

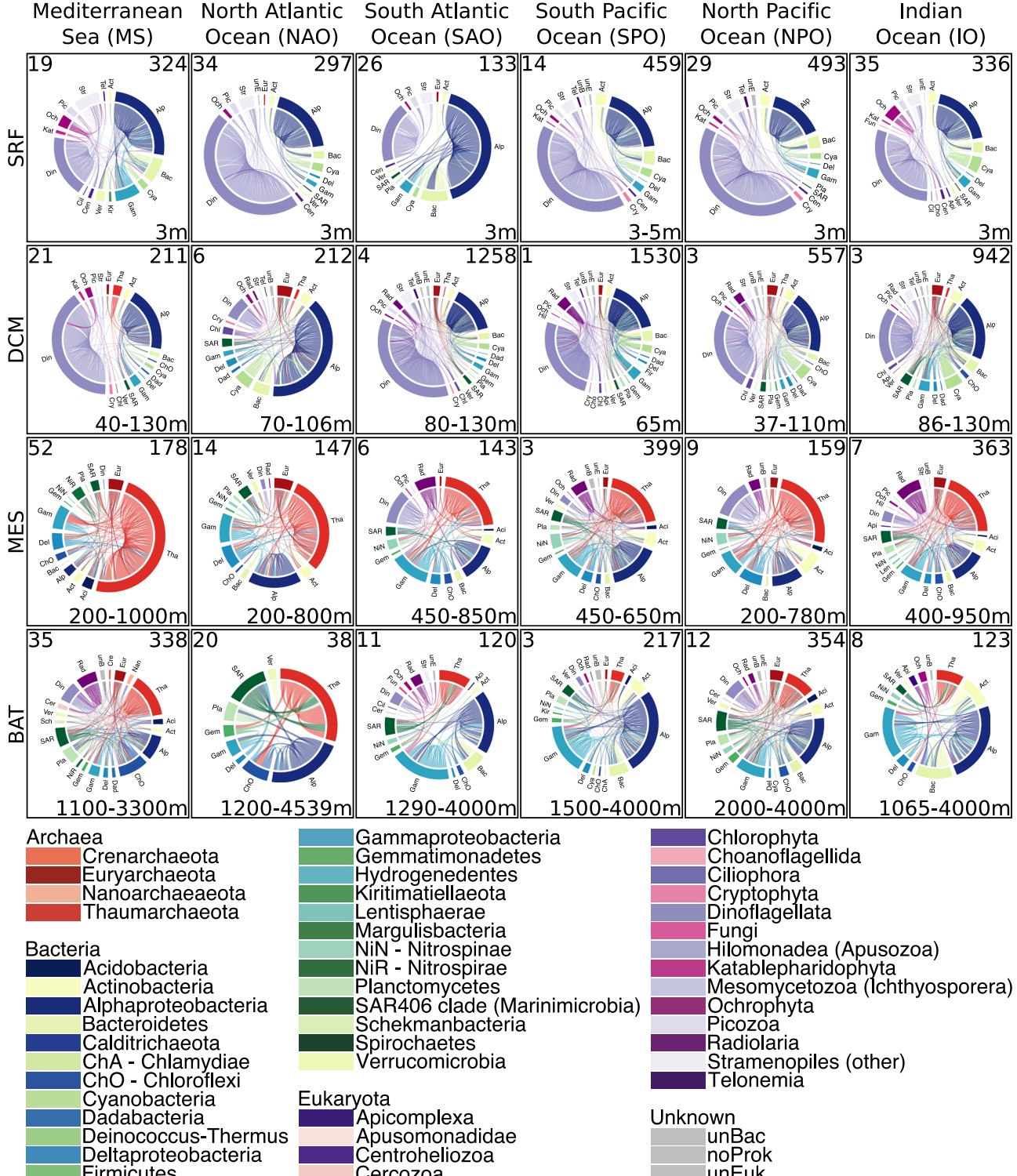

**Fig. 3 | Taxonomic profiles of highly prevalent associations for each region and depth layer.** If an association appears in more than 70% of the subnetworks, it is classified as highly prevalent. Rows indicate the four depth layers: surface (SRF), DCM, mesopelagic (MES), and bathypelagic (BAT). The number of samples appears in the upper left corner, the number of edges in the upper right corner, and the depth range in the lower right corner (in m below surface). The nodes (outer chord) are microbial taxa (ASVs) grouped by taxonomic rank (indicated with colors). The edges (connections inside the chord circle) represent the associations between the ASVs. Note how the proportion of associations changes across the vertical and horizontal dimensions of the ocean. Source data are provided in the GitHub/Zenodo[37] repositories (sections 02_NetworkConstruction and 04_Prevalence; see Data Availability).

addition, most surface associations disappeared with depth in the five ocean basins and MS (Fig. 5), suggesting that most surface ocean interactions among the picoplankton are not transferred to the deep sea, despite the vertical dispersal of various taxa[12]. Specifically, we observed that most deep ocean ASVs already appeared in the upper layers (Supplementary Fig. 1), in agreement with previous work that has shown that a large proportion of deep-sea microbial taxa, including those from small size-fractions, are also found in surface waters,

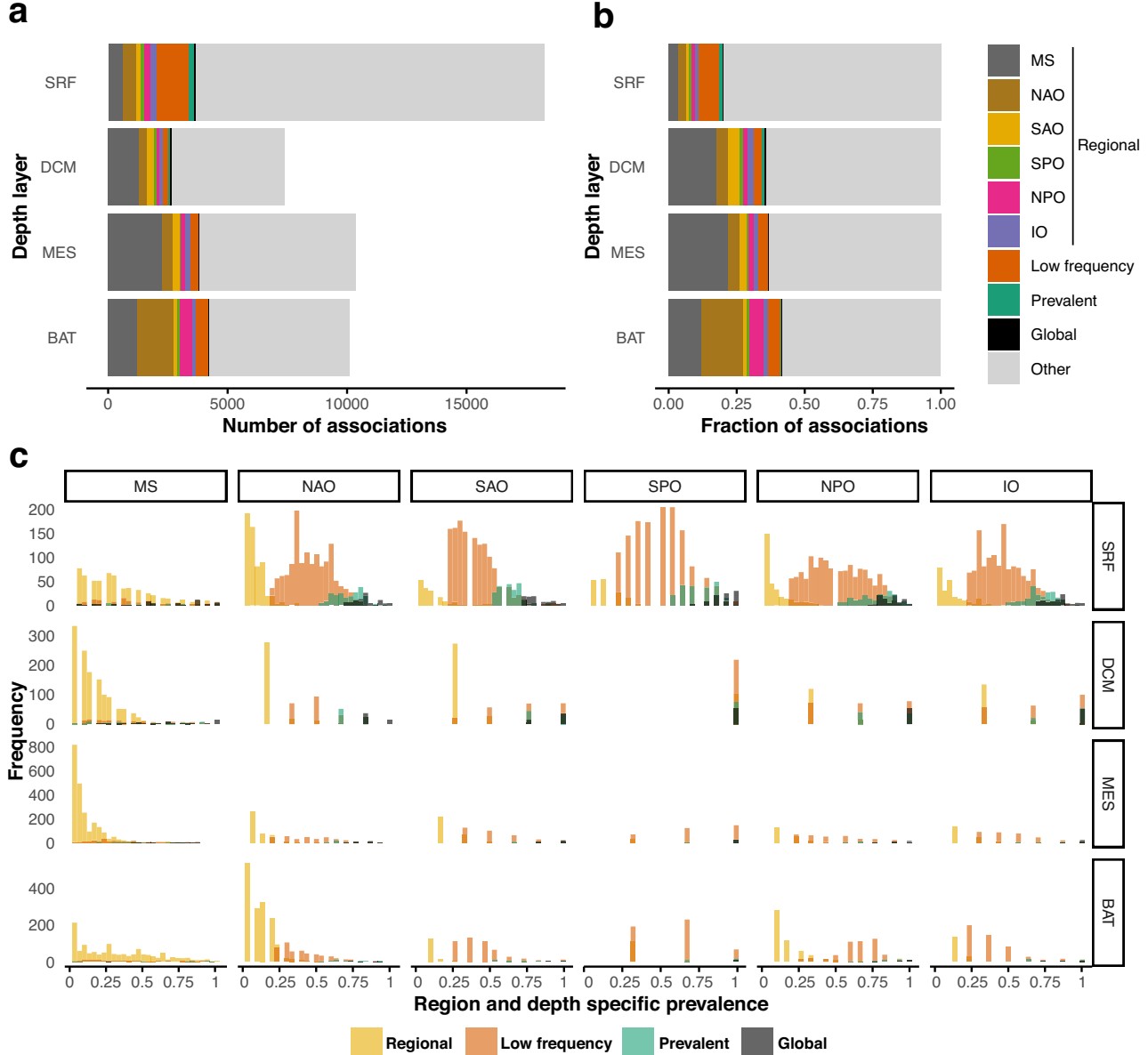

**Fig. 4 | Distribution of potential interactions according to their spatial classification.** We classified association into global (>70% prevalence, not considering the MS), prevalent (≤70% and >50%, not considering the MS), low-frequency (≤50% and >20%, not considering the MS), regional, and other. Regional associations are assigned to one of six ocean regions (five ocean basins and the Mediterranean Sea). The number (**a**) and fraction (**b**) of each type of association are shown for each depth layer: surface (SRF) and DCM (epipelagic), mesopelagic (MES), and bathypelagic (BAT). The color indicates the type of classification. The associations have been classified into five types based on their prevalence in each region. The prevalence of associations is shown in (**c**). For instance, global associations have a prevalence above 70% in each region (not considering the MS). Regional associations are present in one region (indicated with yellow with mainly low prevalence >0%) and absent in all other regions (0% prevalence not shown in graph). Source data are provided in the GitHub/Zenodo[37] repositories (section 05_ClassifyingAssociations; see Data Availability).

and their presence in the deep sea is putatively related to sinking particles[12].

**Environmental gradients seem to shape microbial networks**
Above, we grouped the sample-specific subnetworks based on regions and depth layers. However, such predefined groupings may introduce a bias to our analysis. Thus, we grouped subnetworks based on similar topology (see Methods) and identified 36 clusters of 5–28 subnetworks (Supplementary Table 3). We found 13 (36.1%) clusters dominated by surface subnetworks: six clusters (100% surface subnetworks) from three to five ocean regions but not the MS, and seven clusters including 55–86% surface networks from two to five ocean regions. In

turn, 11 clusters were dominated by other layers: two DCM (64–90%), five mesopelagic (62–83%), and four bathypelagic-dominated clusters (60–69%). Nine of these 11 clusters combined different regions except for one mesopelagic and one bathypelagic-dominated cluster representing the MS (Supplementary Table 3) exclusively. Furthermore, we found 11 clusters containing exclusively or mainly MS subnetworks in contrast to only one cluster dominated by an ocean basin (NAO).

Next, we built a more comprehensive representation of network similarities between subnetworks via a minimal spanning tree (MST, see Methods). The depth layers, ocean regions, location of clusters, and environmental variables were projected onto the MST (Fig. 6). Most surface subnetworks were centrally located, while subnetworks

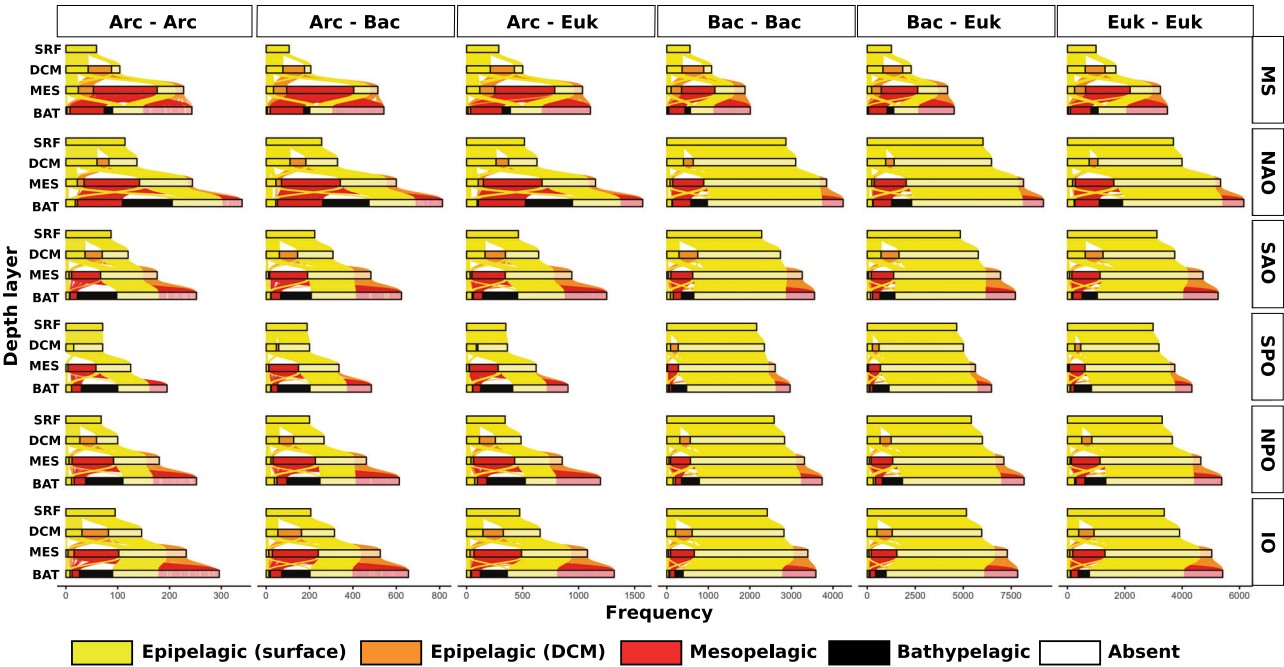

**Fig. 5 | Potential microbial interactions across depth layers.** For each region and taxonomic domain, we color associations based on when they first appeared: surface (S, yellow), DCM (D, orange), mesopelagic (M, red), and bathypelagic (B, black). The SRF bar contains the associations that appeared on the surface. If they also appeared in the DCM, they are listed on the left box of the DCM bar. However, if they were not found in the DCM layer, i.e., they were absent, they appear on the right transparent box of the bar. That is, absent ASVs are grouped in the transparent box at the end of the DCM, MES, and BAT bars. Columns show associations between archaea (Arc), bacteria (Bac), and eukaryotes (Euk). Note that few associations were present throughout the water column within a region and that most associations from the meso- and bathypelagic did not appear in the upper layers except for the MS and NAO. Source data are provided in the GitHub/Zenodo[37] repositories (section 06_VerticalConnectivity; see Data Availability).

from other depths appeared in different MST areas (Fig. 6a). Most MS subnetworks were located in a specific branch of the MST. At the same time, the five ocean basins were mixed (Fig. 6b), indicating homogeneity and connectivity within oceans but network-based differences between the ocean and the MS subnetworks. As expected, networks of the same cluster appear mostly connected in the MST (Fig. 6c). Moreover, subnetworks in the MST tended to connect to subnetworks from the same depth layer or similar environmental conditions (Fig. 6a, d). Overall, our results suggest a strong influence of environmental gradients and, to some extent, geography in shaping microbial network topology in the ocean (Fig. 6a, b, d), as previously observed in epipelagic communities at the global scale[29].

## Discussion

A current challenge is to predict the effects of global change on marine food webs and the overall impact on ecosystem function. Understanding the structure and biogeography of the ocean interactome is an essential step in that direction. Our approach allowed us to investigate the main patterns in the tropical and subtropical global ocean interactome, considering the vertical and horizontal dimensions. Thus, compared to previous works that have focused on the surface layer[27,29], a critical, innovative aspect of this study is the analysis of the deep ocean interactome. Specifically, a key novel contribution of our work is analyzing how the ocean interactome changes with depth across the tropical and subtropical global ocean and the Mediterranean Sea. Even though previous works have already shown how microbial communities change with depth[12,15,39,41], our work goes further by showing how potential microbial interactions and derived network topologies change with depth. Based on 397 samples, our global network contained 5448 nodes and 29,118 edges. A total of 28,178 (96.8%) edges were positive, while 940 (3.2%) were negative. More extensive networks have been inferred from surface-ocean global samples (9169 nodes and 92,633 edges from 313 samples by ref. 27 and 20,810 nodes

and 86,026 edges from 575 samples by ref. 29). Yet, these studies used seven organismal size fractions and two depths (Surface and DCM), while we targeted a single size fraction (the pico-plankton) and four depths (Surface, DCM, Meso- and Bathypelagic). Given that size fractions can recover a larger amount of taxa, the differences in network size between ours and the mentioned studies are not surprising. However, despite these differences, our results and those from the mentioned studies recovered, for the most part, positive associations (between 72 and 98%). This suggests that specific biotic interactions, such as syntrophy or symbiosis, are more prevalent than others, as indicated by temporal network analyses[26].

Positive interactions may underpin the functioning of the ocean microbiome, which could have important implications for ecosystem stability, given that many positive associations could destabilize communities due to positive feedback between species[42]. Then, a decrease or increase in the abundance of one species may pull others with it, leading to a cascade effect that will be propagated through the network. Alternatively, it is also possible that the sampling design and methodological approach missed the majority of negative, or weak interactions (defined here as those associations between microbial taxa that exhibit subtle or low-magnitude correlations)[26]. For example, plummeting species abundances between stations or samples could prevent establishing significant negative correlations. Weak interactions and competition (or other negative interactions) are essential for community stability, as networks including these interactions are less prone to destabilizing cascade effects[42–47]. Considering that the horizontal turnover of the abundant taxa populating the ocean microbiome is usually gradual rather than drastic[5,8], stabilizing mechanisms such as weak interactions and competition are expected to be common. In addition, recent work has shown that networks' vulnerability to global change can differ across ocean regions, being particularly high in the Arctic[29]. Therefore, investigating interactions that stabilize networks or make them more resilient to disturbances is of particular

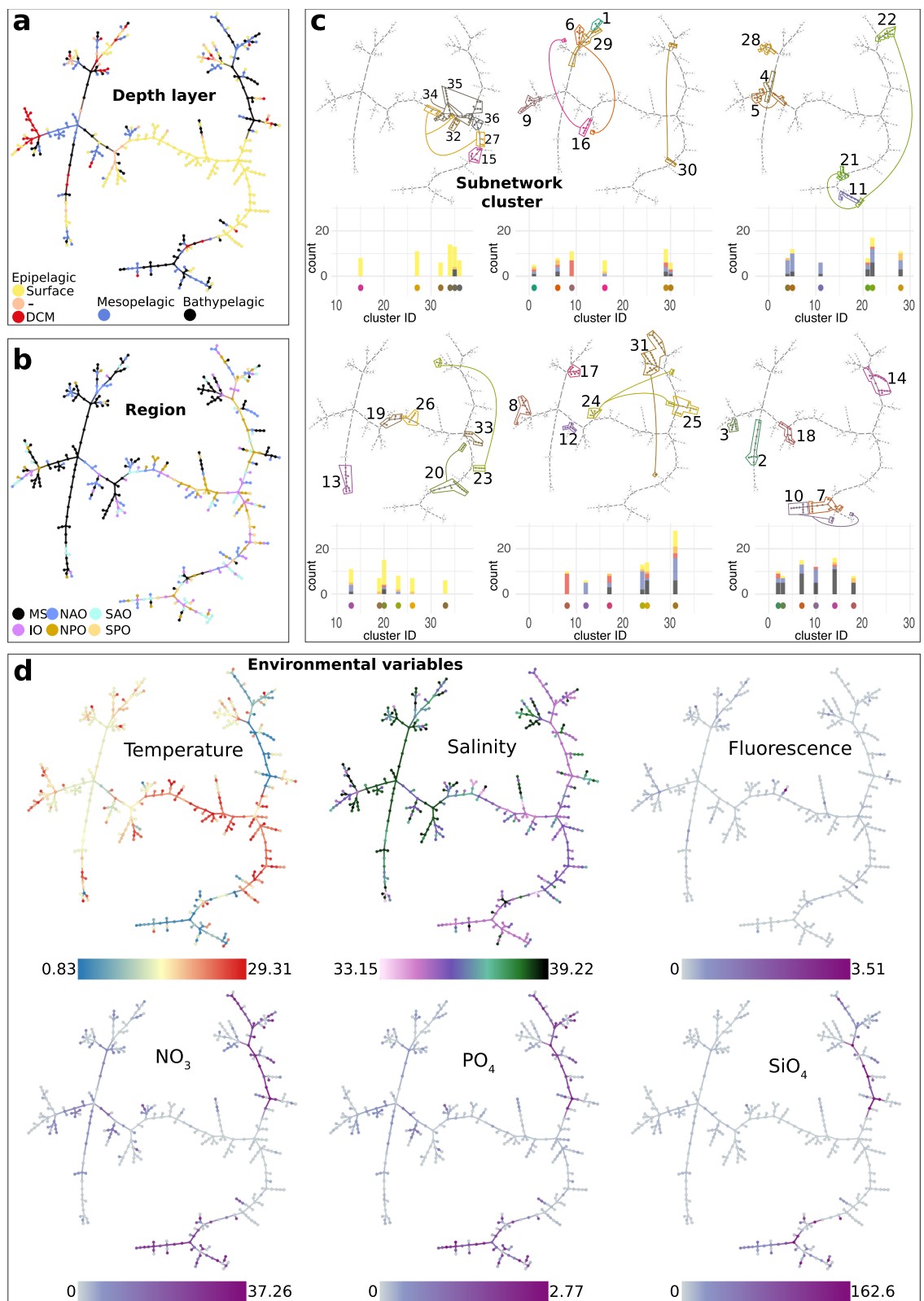

interest, and future work should implement sampling designs and analytical approaches that can better characterize them. For example, high-frequency sampling campaigns including multiple replicates, microcosms and manipulative field experiments, and metabolic modeling coupled to metabolomics[48].

In both our work and that of others using networks based on the 16S and 18S rRNA gene markers (e.g., refs. 26,27,29,36), it is assumed

that the taxonomy may capture functional traits involved in the interactions through effects in microbial abundances (co-occurrences or co-exclusions). A specific microbial interaction must have enough specificity to be detected over space or time as a co-occurrence or co-exclusion pattern. However, promiscuous interactions due to functional redundancy (for example, one microbe that can interact with several different ones that are ecologically equivalent) may not have

**Fig. 6 | Similarity of sub-networks based on topology.** Minimal Spanning Tree (MST) based on dissimilarities in sub-network topologies. Each of the 397 nodes in the MST represents a unique subnetwork derived from specific samples. The 396 edges in the MST represent the dissimilarity between sub-network pairs (samples). Edge weights are assigned based on the network-dissimilarity score. MST aims to connect all 397 nodes to one connected tree under the condition that the 396 edges have the lowest sum of scores. Note there may be more than one solution for an MST. The MST allows simplifying the main similarity patterns between subnetworks, i.e., nodes closer to each other indicate that sub-networks are more similar in topology and vice-versa. The shown MST allows a broad comparison of subnetworks based on their similarities in interconnection patterns or topologies. The similarity in topology suggests that similar ecological processes, environmental conditions, geographic features, or a mix of them have influenced the microbial communities and their potential interactions in these subnetworks. We have mapped on the MST geographic, oceanographic, and environmental variables. If nodes clustering close to each other in the MST share similar magnitudes in some of these variables, then such variables could be influencing the topology of the sub-networks (e.g., in (**a**), multiple sub-networks from the surface ocean are grouping, indicating that they tend to be more similar among themselves than to sub-networks from other depth layers). Nodes are colored according to (**a**) the sample's depth layer, (**b**) the sample's ocean region, (**c**) the subnetworks cluster (see Supplementary Table 3), and (**d**) selected environmental variables. In **c** the bar plots indicate the different depth layers, colored as in (**a**), within each cluster. For each cluster, we show how many subnetworks belong to Surface, DCM, Mesopelagic, and Bathypelagic. The cluster is indicated by color (below bars and on top of the MST) and cluster number (x-axis). The depth of samples (Surface, DCM, Mesopelagic, and Bathypelagic) within clusters is indicated by the colors in the bars (see color code in **a**) and the number of different depth layers in each cluster by the bars' height (y-axis). Source data are provided in the GitHub/Zenodo[37] repositories (section 07_NetworkSimilarity; see Data Availability).

enough abundance signal and may not be detected. Despite this and other limitations of the association networks, as discussed in detail elsewhere[49], the signal we obtain using microbial abundances based on rRNA gene markers can reflect strong spatial associations with substantial chances to represent ecological interactions. Thus, even though its known limitations, association networks are still one of the best tools available to address the vast complexity of natural microbial interactions. Other approaches, such as metabolic modeling and metabolomics[48], are promising in detecting the role of functional redundancy in microbial interaction networks. Yet, these approaches are starting to be implemented in environmental studies, and more development is needed to apply them to complex marine communities over large spatiotemporal scales.

By using an innovative approach to determine sample-specific interactomes, we identified global (i.e., present in all regions with >70% prevalence for a given depth layer) and regional pelagic microbial associations across the oceans' vertical and horizontal dimensions. We found few global associations, indicating a potentially small core interactome in the tropical and subtropical global ocean and the Mediterranean Sea within each depth layer. Our results agree with those from Chaffron et al.[29], which showed that global or widespread associations were a minority and that most associations were community-specific. These results align with those reporting a relatively small number (between 0.3-1%) of widespread microbial taxa in the surface ocean[5], given that more geographically ubiquitous taxa could lead to more widespread interactions. In turn, these results contrast with those of ref. 27, who found a large proportion of widespread associations in the global sunlit ocean. Most likely, this reflects differences in the used approaches and datasets, as our work and that of ref. 29 implement a recently developed network construction tool that infers direct associations from heterogeneous microbial abundance datasets (FlashWeave[50]), and, in addition, both works analyzed sample-specific networks. The previous analyses were not implemented by ref. 27, and this could be a reason for the observed differences. Both mentioned studies were based on TARA Oceans datasets and have used OTUs or miTags[51], while here, we have used ASVs with higher taxonomic resolution[52], which could also contribute to differences between the results.

Recently, Milke et al.[53] have reported that prokaryotic communities in the epipelagic Pacific, Atlantic, and southern Indian Oceans and the Mediterranean Sea are structured into modules of co-occurring taxa with specific distributions and environmental preferences. In contrast, our work extends beyond the epipelagic zone to include deeper layers and microbial eukaryotes, emphasizing the dynamic nature of potential interactions across the horizontal and vertical dimensions of the tropical and subtropical global ocean and the Mediterranean Sea. Thus, while Milke *et al.*'s work underscores the importance of specific co-occurring taxa across vast oceanic regions, our analyses focus on the change of potential interactions with depth and ocean basin. Both works complement each other by indicating that marine microbial communities include both stable, tightly-knit associations as well as associations that are more idiosyncratic or that can change over space and time[36].

Ocean currents are generally stronger, and the variability in environmental conditions is greater in surface waters compared to the deep ocean[54]. This could affect the relative importance of selection and dispersal in structuring microbial communities and, consequently, association networks. Within each oceanic region, we found less highly prevalent associations in the bathypelagic zone of the tropical and subtropical global ocean (pointing to a smaller regional core) than in the upper layers, except for the NPO, which had less highly prevalent associations in the meso- than in the bathypelagic. In agreement, we found more regional associations in the bathypelagic than in the upper layers. This may reflect a higher dispersal limitation in deep ocean regions due to slow currents, water masses[54], straits, and seamounts[55]. Consistently, using the same picoplankton dataset, we observed that selection tends to decrease while dispersal limitation tends to increase with depth in the tropical and subtropical global ocean and the Mediterranean Sea[38]. Other studies have also reported a higher dispersal limitation with depth in specific ocean regions[32]. A recent study found increasing differences in picoplankton community composition with depth when comparing the surface vs. the deep layer of different ocean basins[39]. Yet, when analyzing the entire dataset encompassing the tropical and subtropical ocean, the previous work found that surface prokaryotic communities were more different across stations than deep counterparts, while surface picoeukaryotic communities were more similar than deep ones. This could be attributed to the differential action of selection and dispersal limitation in prokaryotes and picoeukaryotes across water layers[38,39].

Even though environmental gradients in the deep ocean are smooth when compared to the surface[54], specific processes could contribute to increasing the complexity and number of niches in the deep ocean, potentially leading to an increase in the number of regional associations. For example, different niches may be associated with the quality and types (labile vs. recalcitrant) of organic matter reaching the deep ocean from the epipelagic zone[30], which is significantly different across oceanic regions[56]. In an exploration of generalists versus specialist prokaryotic metagenome-assembled genomes in the Arctic Ocean, most specialists were linked to meso-pelagic samples, indicating that their distribution was uneven across depth layers[57].

We sought to understand the impact of environmental gradients and geographic factors on the topology (that is, patterns of network connections) of microbial networks in the tropical and subtropical global ocean and the Mediterranean Sea across depths. For that, we clustered subnetworks based on topology similarity, resulting in 36 clusters with diverse compositions, and built a minimal spanning tree (MST) to display subnetwork similarity. The MST revealed a tendency

for subnetworks to connect with other subnetworks from the same depth layer or similar environmental conditions. This highlights the significant role of environmental gradients and geographic factors in determining the topology of microbial networks in the ocean. While the broad action of ecological processes (that is, selection, dispersal, and drift)[4] on microbial interactions still needs to be investigated, our results suggest that selection and dispersal limitation could have a relevant role in shaping the deep-sea interactome. These findings align with earlier research on global epipelagic networks[27,29] and a previous work where we found that prokaryotes inhabiting locations in the tropical and subtropical surface ocean featuring similar temperatures tend to co-occur[5]. In the last study, we used the TINA index[58], which aims to quantify the similarity between two communities as the average interaction strength between all taxa observed in them, while here, we increase the resolution of the environmental analyses by comparing the actual topologies of networks using graphlets. Altogether, and regardless of the analysis type, multiple pieces of evidence point to environmental heterogeneity having a substantial effect in shaping the topology of association networks in the ocean.

Environmental factors, such as temperature, nutrient availability, salinity, and light, among others, have been shown to influence the distribution, abundance, and activity of marine microorganisms[5-7,38,39,59,60]. Results from the MST point to the importance of environmental gradients, exerting selection, in shaping the topology of microbial networks across marine ecosystems. Analyses of the spatial and temporal variation of marine microbial networks have further stressed the crucial role of environmental factors in shaping network architecture[29,36,61]. In a study of the surface global-ocean interactome Chaffron et al. [29] found few direct associations between taxa and environmental variables, similar to our findings when investigating a marine-coastal interactome over ten years[26]. For interactomes where positive associations predominate, such as the two previous works plus others[24,27,62], this suggests that environmental variables may significantly influence a number of species. These would then pull the others, a process facilitated by positive associations, generating cascade effects and specific network dynamics[42], as we have previously suggested for a marine coastal interactome[26]. It is worth reminding that we have removed indirect edges that reflect similar environmental preferences and not potential interactions by using FlashWeave coupled with EnDED (see Methods).

The observed differences in network topology across distinct but environmentally similar oceanic regions or basins suggest that regional processes also play a role in determining network topology. Different dispersal rates between ocean regions may be responsible for a substantial fraction of the regional effects. This aligns with recent results that found compelling evidence that ocean currents exert a significant basin-scale influence on microbial plankton biogeography[63]. Furthermore, stochastic changes in community composition, or drift, could also underpin regional effects. Recent studies found evidence that dispersal, drift, and selection change with depth, among ocean basins, and between basins and the Mediterranean Sea[5,9,38]. This variability in the relative importance of the ecological process shaping communities may explain, to a certain extent, the changes in network topologies across regions. Other unmeasured processes could also play a role, such as the promiscuity of interactions (that is, the possibility of one microbe establishing ecological interactions with the same or different partners across ocean regions[61]).

Vertical connectivity in the ocean microbiome is partially modulated by surface productivity through sinking particles[12,41,64]. An analysis of eight stations distributed across the Atlantic, Pacific, and Indian oceans (including four depths: Surface, DCM, meso- and bathypelagic) indicated that bathypelagic communities comprise both endemic taxa as well as surface-related taxa arriving via sinking particles[12]. Ruiz-González et al.[41] identified both components (i.e., surface-related and

deep-endemic) and the dominating phylogenetic groups. While Thaumarchaeota, Deltaproteobacteria, OM190 (Planctomycetes), and Planctomycetacia (Planctomycetes) dominated the endemic bathypelagic communities, Actinobacteria, Alphaproteobacteria, Gammaproteobacteria, and Flavobacteriia (Bacteroidetes) dominated the surface-related taxa in the bathypelagic zone[41]. We found association partners for each of these dominating phylogenetic groups within each investigated association type: highly prevalent, regional, global, prevalent, and low-frequency associations. While ASVs belonging to these taxonomic groups were present throughout the water column, specific associations were observed mainly in the mesopelagic and bathypelagic zones. This suggests specific interactions between endemic deep-sea taxa, in agreement with the hypothesis indicating high niche partitioning and more specialist taxa in the deep ocean[65,66]. Accordingly, a recent study found a remarkable taxonomic and functional novelty in the deep ocean after analyzing 58 microbial metagenomes from a global deep-sea survey, unveiling ~68% archaeal and ~58% bacterial novel species[67].

Little is known about the distribution of microbial interactions across the water column. Associations found along the entire water column could point to microbes interacting across all water layers or interacting microbes that sink together[68]. We found that associations present across all layers were limited, pointing to a heterogeneous distribution of interactions in the water column. Given that we targeted the picoplankton, the associated taxa found in the entire water column may represent non-physical interactions occurring in all water layers instead of interactions occurring in sinking particles[68]. A fraction of the associations observed only in the deep ocean may correspond to microbial consortia degrading sinking particles or taxa that might have detached from sinking particles, i.e., dual lifestyle taxa, as observed by ref. [69]. Our results suggest that most microbial interactions change across the water column while a few are maintained. Furthermore, some microorganisms may change their interaction partners across the water column. Changes in microbial interactions with depth could also be linked to ecological successions in sinking particles[70].

Network topologies changed with depth. Deep-sea networks were more clustered (higher transitivity) and had higher average path lengths (average number of steps (or "edges") that must be traversed to go from one node to any other node in the network), displayed stronger associations, and lower degree assortativity (nodes are less likely to associate with other nodes with similar degree) than surface networks. These topological changes may have multiple ecological implications. An increased clustering suggests more specialized or tightly-knit ecological relationships in the deep ocean, pointing to niche-specific microbes and interactions that could be potentially vulnerable to ocean change. For example, if a key species were to be lost, it could have a cascading effect on the entire community[42]. This aligns with a hypothesis indicating a high niche partitioning and more specialist taxa in the deep ocean[65,66]. A higher average path length indicates a more complex or fragmented network structure that could affect, for example, the transfer of carbon and energy through the community[71]. Stronger positive associations among deep-sea microbes imply more stable and persistent interactions, likely influenced by the stable environmental conditions in the deep ocean[54]. Yet, these interaction types could make communities more vulnerable to environmental changes via cascade effects[42]. Lastly, the lower assortativity in deep-sea networks implies a more heterogeneous structure, with microbes potentially interacting across a broader range of functional roles, metabolisms, or ecological niches. Overall, microbial networks in the deep sea seem more complex than surface counterparts in the tropical and subtropical ocean. A recent study investigating microbial networks in the western Pacific Ocean from 0 to 2000 m also found clear differences between aphotic and photic networks[32]. Furthermore, in agreement with our results, the previous study found that deep-sea networks had fewer edges and a higher average path

length, pointing to looser connectivity than surface networks. Yet, contrary to our results, the study[32] reports a higher clustering (transitivity) of photic networks compared to aphotic. This discrepancy could be linked to different sampling and network construction strategies, differences in the delineation of depth layers, total sampling depth, or the specific region that was investigated. In any case, our results and those from the mentioned work indicate a clear change in the ocean interactome as we dive into the deep ocean, which aligns with the observed vertical community changes[12,15,39], and which could imply different effects in the functioning of surface and deep-sea microbial ecosystems[71].

On average, mesopelagic subnetworks displayed the lowest network connectivity (determined via edge density) across most regions. We found the most robust associations among both meso- and bathypelagic subnetworks. Moreover, we found the highest clustering (transitivity) in the meso- and bathypelagic zones (relatively colder waters) compared to the epipelagic zone (warmer waters). Similarly, another global-scale study[29], concentrating on the epipelagic zone and including polar waters, found higher edge density, association strength, and clustering in polar waters compared to warmer waters. These results suggest that either microorganisms interact more in colder environments or that their recurrence is higher due to a higher environmental selection exerted by low temperatures. Alternatively, limited resources (primarily nutrients) on the surface versus the deep tropical and subtropical ocean may prevent the establishment of specific microbial interactions in surface waters. Furthermore, environmental stability in the deep sea may have led to high niche partitioning[65,66], which could have promoted the establishment of meso- and bathypelagic interactions.

Through quantifying regional associations, our results indicated distinct associations in the MS, where most regional associations were observed compared to the ocean basins. Similar results were reported by ref. 27, who found that two-thirds of the epipelagic regional associations originated from the MS compared to other ocean basins. The MS has unique features compared to the open ocean, such as higher temperature and salinity in deep waters and a west-to-east gradient of decreasing nutrient concentration and increasing salinity in surface waters[59,72], which could have favored the establishment of local interactions. Furthermore, the Mediterranean Sea is a hotspot of multicellular biodiversity and endemic species, several of which may have emerged during periods of isolation from the global ocean[73,74]. Despite being less studied than animals and plants, there are also reports of putatively endemic MS microorganisms, such as specific SAR11[75] or monophyletic ecotypes of bathypelagic *Crenarchaeota*[76]. Thus, part of the recovered MS associations could reflect endemic interactions involving endemic and non-endemic taxa. Our findings from the Mediterranean Sea suggest that other enclosed or semi-enclosed seas, such as the Baltic Sea, Black Sea, Caspian Sea, and Red Sea, could harbor endemic or regional microbial interactions. In any case, potentially endemic microbial taxa should be investigated at the genome level, given that the 16S or 18S rRNA gene may not reflect fine-grained differences[77,78]. Furthermore, we found a substantial number of regional associations in the NAO compared to other ocean basins, contrasting with the NAO having the lowest number of regional associations in a previous epipelagic network[27]. The previous study used different samples and multiple microbial size fractions, which could explain the differences between both studies. How many of the regional associations that we detected represent endemic interactions needs further investigation. Even though our dataset is one of the largest generated so far for the tropical and subtropical global ocean and the Mediterranean Sea, it includes different sampling efforts for regions and depth layers. This may have introduced biases in the detected regional or global associations. Furthermore, different methodologies should be tested, as network construction tools, analytical thresholds, sequencing depth, and sampling design could all

influence the amount of putatively endemic interactions that are detected. Despite these limitations, we consider that our work provides important insights into the amount of regional and cosmopolitan putative interactions in the tropical and subtropical global ocean and the Mediterranean Sea.

To conclude, we have analyzed the spatial distribution of potential microbial interactions in the tropical and subtropical global ocean and the Mediterranean Sea, considering archaea, bacteria, and picoeukaryotes from surface to bottom waters. Thus, our work significantly expands previous efforts that focused on analyzing the microbial interactome of upper water layers of the global ocean[27,29]. We have used an innovative approach, sample-specific networks, that allowed us to analyze the change of networks across locations and determine global and regional microbial associations across water layers. Therefore, our work contributes to understanding the dynamics of the ocean interactome, still a developing research field[79]. We found few global associations for the horizontal dimension of the ocean. In turn, 11-36% of the associations were regional and limited to specific depth layers. For the vertical dimension, our results indicate that associations change across the water column and that they may have specific depth distributions. Previous studies have investigated the horizontal and vertical turnover of the ocean microbiome (that is, community composition)[5,6,8,12,15]. Our results expand this knowledge by indicating how potential interactions may change over the vertical and horizontal dimensions of the ocean. Furthermore, our results contribute to understanding the biogeography of potential interactions and provide hints on the links between different network topologies and ecosystem functioning, which is relevant in the context of global change[80]. Finally, the identified widespread microbial associations, which could be important for ocean ecosystem function across multiple locations, could be the initial target of future monitoring or conservation efforts to preserve the ocean microbiome and interactome[81].

## Methods
### Dataset
Samples originated from two expeditions, Malaspina-2010[82] and Hotmix[83]. The former was onboard the R/V Hespérides, and most ocean basins were sampled between December 2010 and July 2011. Malaspina samples included i) *MalaSurf*, surface samples[5,40], ii) *MalaVP*, vertical profiles[15], and iii) *MalaDeep*, deep-sea samples[84–86]. For the Hotmix expedition, sampling took place onboard the R/V Sarmiento de Gamboa between 27th April and 29th May 2014 and represented a quasi-synoptic transect across the MS and the adjacent North-East of the NAO. See details in Table 2.

DNA extractions are indicated in the publications associated with each dataset (Table 2). The 16S and 18S rRNA genes were amplified and sequenced. PCR amplification and sequencing of *MalaSurf, MalaVP* (18S), and *Hotmix* (16S) are indicated in the publications associated with each dataset in Table 2. *MalaVP* (16S) and *Hotmix* (18S) were PCR-amplified and sequenced following the same approach as in ref. 5. The DNA from *MalaDeep* samples was extracted as indicated in refs. 84,85 and re-sequenced at Genoscope (France) with the primers indicated below. *MalaSurf, MalaVP*, and *Hotmix* datasets were sequenced at RTL Genomics (Texas, USA). Publicly available datasets from the global campaign TARA Oceans[27,29] were not considered due to differences in the used methodologies to obtain the data (e.g., miTags vs. amplicons[51] and different marker regions of the rRNA gene).

We used the same amplification primers for all samples. For the 16S, we amplified the V4-V5 hypervariable region using the primers 515F-Y and 926R[87]. For the 18S, we amplified the V4 hypervariable region with the primers TAReukFWD1 and TAReukREV3[88]. See more details in refs. 5,38. Amplicons were sequenced in *Illumina* MiSeq or HiSeq2500 platforms (2 × 250 or 2 × 300 bp reads). Operational Taxonomic Units were delineated as Amplicon Sequence Variants

## Table 2 | Used datasets

| Dataset | Samples used for analysis | Stations | Depth range (m) | Water samples | Size Fraction (µm) | 16S | 18S | Reference | ENA accession number |
|---|---|---|---|---|---|---|---|---|---|
| Malaspina | | | | | | | | | |
| MalaSurf | 122 | 120 | 3 | 122 | 0.2-3 | 122 | 124 | [5,40] | PRJEB23913 [18S rRNA genes], PRJEB25224 [16S rRNA genes] |
| MalaVP | 83 | 13 | 3-4000 | 91 | 0.2-3 | 91 | 83 | [15] & This study | PRJEB23771 [18S rRNA genes], PRJEB45015 [16S rRNA genes] |
| MalaDeep (Prok[a]) | 13 | 30 | ~4000 | 60 | 0.2-0.8 | 41 | - | [86] | PRJEB45011 |
| MalaDeep (Euk[a]) | 13 | 27 | 2400-4000 | 27 | 0.8-20 | - | 82 | This study | PRJEB45014 |
| Hotmix | 179 | 29 | 3-4539 | 188 | 0.2-3 | 188 | 179 | [59] & This study | PRJEB44683 [18S rRNA genes], PRJEB44474 [16S rRNA genes] |

We required each sampling point to provide data for both eukaryotes and prokaryotes, which resulted in 397 samples. This condition allowed only 13 MalaDeep samples. 16S and 18S refer to sequenced samples.

[a]*Prok* prokaryotes, *Euk* eukaryotes.

(ASVs) using DADA2 v1.20[52], running each dataset separately before merging the results. ASVs were assigned taxonomy using SILVA[89], v132, for prokaryotes, and PR2[90] v4.11.1, for eukaryotes. ASVs corresponding to Plastids, Mitochondria, Metazoa, and Plantae were removed. Only samples with at least 2000 reads were kept. The dataset contained several *MalaDeep* replicates, which were merged, and two filter size fractions. Given the cell sizes of prokaryotes versus microeukaryotes, we used the smallest size-fraction (0.2–0.8 µm) for prokaryotes and the larger one (0.8–20 µm) for microbial eukaryotes. The other three datasets considered the 0.2-3 µm size fraction only. Additionally, we required that samples had eukaryotic and prokaryotic data, resulting in 397 samples for downstream analysis: 122 *MalaSurf*, 83 *MalaVP*, 13 *MalaDeep*, and 179 *Hotmix*. We separated the samples into epipelagic, mesopelagic, and bathypelagic zone (Fig. 1). Furthermore, we separated most epipelagic zone samples into surface layer and deep-chlorophyll maximum (DCM) layer, but 18 MS and 4 NAO samples belonged to neither. We also considered environmental variables: Temperature (2 missing values = mv), salinity (2 mv), fluorescence (3 mv), and inorganic nutrients $NO_3^-$ (36 mv), $PO_4^{3-}$ (38 mv), and $SiO_2$ (37 mv), which were measured as indicated elsewhere[5,15,59]. In specific samples, missing nutrient data were estimated from the World Ocean Database[91].

### Single static network

We constructed the single static network in four steps. First, we prepared the data for network construction. We excluded rare microorganisms by keeping ASVs with a sequence abundance sum above 100 reads across all samples and appearing in at least 20 samples (>5% of the dataset). The latter condition removed eukaryotes only appearing in the 13 *MalaDeep* eukaryotic samples of the 0.8–20 µm size fraction. This filtering step left 2922 eukaryotic ASVs, representing 79.8% of the original eukaryotic reads, and 2535 prokaryotic ASVs, representing 84.8% of the original prokaryotic reads. To control for data compositionality[92], we applied a centered-log-ratio transformation separately to the prokaryotic and eukaryotic tables before merging them.

Second, we inferred a (preliminary) network using FlashWeave v0.18.0[50], based on 5457 ASVs selecting the options "heterogeneous" and "sensitive". FlashWeave was chosen as it can handle sparse datasets like ours, taking zeros into account and avoiding spurious correlations between ASVs that share many zeros. This initial network had 5457 nodes and 31,966 edges, 30657 (95.9%) positive and 1309 (4.1%) negative.

Third, we aimed at removing environmentally-driven edges. FlashWeave can detect indirect edges and can also consider metadata such as environmental variables, but currently, it does not support

missing data. Thus, we applied EnDED v1.0.1 (Environmentally Driven Edge Detection)[93], a method suitable for large-scale spatial data designed to identify which links between microorganisms in an association network are environmentally driven. The program implements four methods: Sign Pattern, Overlap, Interaction Information, and Data Processing Inequality[93]. EnDED can use these methods individually or in combination to better predict environmentally driven associations in microbial networks. In EnDED, we combined the methods Interaction Information (with 0.05 significance threshold and 10000 iterations) and Data Processing Inequality as done previously via artificially-inserted edges to connect all microbial nodes to the six environmental parameters[36]. Although EnDED can handle missing environmental data when calculating intermediate values relating ASV and environmental factors, it would compute intermediate values for microbial edges using all samples. Thus, to avoid a possible bias and speed up the calculation process, we applied EnDED individually for each environmental factor, using only the samples containing values for the specific environmental factor. We detected and removed potential environmentally-driven edges due to nutrients (4.9% $NO_3^-$, 4.2% $PO_4^{3-}$, 2.0% $SiO_2$), temperature (1.9%), salinity (0.2%), and Fluorescence (0.01%) (Supplementary Table 4).

Fourth, we removed isolated nodes, i.e., nodes without any edge. The resulting network represented the single static network in our study. It contained 5448 nodes and 29118 edges; 28178 (96.8%) were positive, and 940 (3.2%) were negative.

### Sample-specific subnetwork

We constructed 397 sample-specific subnetworks. Each subnetwork represented one sample and was derived from the single static network, i.e., a subnetwork contained nodes and edges present in the single static network but not vice versa. First, we require that an edge must be present in the single static network. Second, an edge can only be present within a subnetwork if both microorganisms associated with the edge have a sequence abundance above zero in the corresponding sample. Third, microorganisms associated need to appear together (intersection) in more than 20% of the samples, in which one or both appear (union) for a specific region and depth.

Formally, consider sample $s_{RL}$ with $R$ being the marine region and $L$ the sample's depth layer. Let $e$ be an association between microorganisms $A$ and $B$. Then, association $e$ is present in the sample-specific subnetwork $N_s$, if

i. $e$ is an association in the single static network,

ii. the microorganisms $A$ and $B$ are present within sample $s$, i.e., the abundances are above zero within that particular sample, and

iii. the association has a region and depth-specific Jaccard index, $J_{RL}$, above 20% (see below).

In addition to these three conditions, a node is present in a sample-specific subnetwork when connected to at least one edge, i.e., we removed isolated nodes.

Regarding the third condition, we determined $J_{RL}$ for each association pair by computing within each region and depth layer, the fraction of samples two microorganisms appeared together (intersection) from the total samples at least one microorganism appears (union). Supplementary Table 5 shows the number of edges using different thresholds. Given the heterogeneity of the dataset within regions and depth layers, we decided to use a low threshold, keeping edges with a Jaccard index above 20% and removing edges below or equal to 20%. The third condition was robust (Supplementary Fig. 5). We tested robustness by randomly drawing a subset of samples from each region and depth combination. The subset contained between 10% and 90% of the original samples. We rounded up decimal numbers to avoid zero sample subsets, e.g., 10% of 7 samples results in a subset of 1 sample. We excluded the DCM of the SPO because it contained only one sample. Next, we recomputed the Jaccard index for the random subset. Lastly, requiring $J > 20\%$, we evaluated the robustness of the third condition (i.e., the association has a region and depth-specific Jaccard index, $J_{RL}$, above 20%), for generating sample-specific subnetworks for each region and depth with sufficient samples. Within each region and depth, the samples were randomly subsampled, containing 10% to 90% of the original set using all samples. We determined the fraction of edges kept in the subsampled set compared to the original set. Specifically, we determined i) how many edges were kept in the random subsamples compared to all samples (that is, only the number of kept edges) and ii) how many edges were kept in the random subset that were also kept when all samples were used (that is, which edges were kept). We repeated the procedure for each region-depth combination 1000 times.

## Spatial recurrence

To determine an association's spatial recurrence, we calculated its prevalence as the fraction of subnetworks in which the association was present. We determined association prevalence across the 397 samples and each region-layer combination. We mapped the scores onto the single static network and visualized them in Gephi[94] v.0.9.2 using the Fruchterman Reingold Layout[95] with a low gravity score of 0.5. We used the region-layer prevalence to determine global and regional associations. We considered an association to be global within a specific depth layer if its prevalence was above 70% in all regions. In turn, a regional association had an association prevalence above 0% within a particular region layer (present, appearing in at least one subnetwork) and 0% within other regions of the same layer (absent, appearing in no subnetwork). We further characterized associations that were neither global nor local. We considered an association prevalent within a specific depth layer if its prevalence was above 50% in all regions. Similarly, associations that appear in a specific depth layer in all regions over 20% are considered low-frequency. Thus, an association can be classified as i) global, ii) regional, iii) prevalent, iv) low-frequency, and v) "other", i.e., associations that have not been classified into the aforementioned categories.

## Network metrics

We considered the *number of nodes* and *edges* and six other network metrics, most of them computed with the *igraph* v1.2.6 R-package[96]. *Edge density* indicating connectivity is computed through the number of actual edges divided by the number of possible edges. The *average path length* is the average length of all shortest paths between nodes in a network. *Transitivity*, indicating how well a network is clustered, is the probability that the nodes' neighbors are connected. *Assortativity* measures if similar nodes tend to be connected, i.e., *assortativity (degree)* is positive if high-degree nodes tend to connect to other high-degree nodes and negative otherwise. Similarly, *assortativity (Euk-Prok)* is positive if eukaryotes tend to connect to other eukaryotes while prokaryotes tend to connect to other prokaryotes. Lastly, we computed the *average positive association strength* as the mean of all positive association scores provided by FlashWeave.

## Similar networks based on network topology

The previous metrics (so-called global network metrics) disregard local structures' complexity, and topological analyses should include local metrics[97], e.g., graphlets[98]. Here, we determined network dissimilarity between each pair of sample-specific subnetworks as proposed in ref. 99, comparing network topology without considering specific ASVs. The network-dissimilarity is a consistently positive distance measurement: 0 if networks are identical, while larger numbers indicate greater dissimilarity.

Next, we constructed a Network Similarity Network (NSN), where each node is a subnetwork, and each node connects with all other nodes, i.e., the NSN was a complete graph. Then, we assigned the network-dissimilarity score as edge weight within the NSN. Finally, we determined the NSN's minimal spanning tree (MST) to simplify the NSN while preserving its main patterns. The MST had 397 nodes and 396 edges. The MST is a backbone with no circular path in which the edges are chosen so that the edge weights sum is minimal and all nodes are connected, i.e., a path exists between any two nodes. We determined the MST using the function *mst* in the igraph package in R[96,100].

Using the network-dissimilarity (distance) matrix, we determined clusters of similar subnetworks using Python 3 scripts. First, we reduced the matrix to ten dimensions using *umap* v0.5.2[101] with the following parameter settings: n_neighbors=3, min_dist=0, n_components=10, random_state=123, and metric='precomputed'. Second, we clustered the subnetworks (represented via ten dimensions) with *hdbscan* v0.8.27[101], setting the parameters to min_samples=3 and min_clusters=5.

## Reporting summary

Further information on research design is available in the Nature Portfolio Reporting Summary linked to this article.

## Data availability

DNA sequence data is publicly available at the European Nucleotide Archive (see details in Table 2). The accession numbers for the different datasets are: *MalaSurf* (PRJEB23913, PRJEB25224), *MalaVP* (PRJEB23771, PRJEB45015), *MalaDeep* (PRJEB45011, PRJEB45014), *Hotmix* (PRJEB44683, PRJEB44474). OTU tables and source data to generate the figures and tables are provided in GitHub (https://github.com/InaMariaDeutschmann/GlobalNetworkMalaspinaHotmix) and Zenodo: https://doi.org/10.5281/zenodo.10230073[37]. The following databases have been used: SILVA v132[89], PR2 v4.11.1[90], and the World Ocean Database 2013[91].

## Code availability

The code for data analysis, including commands to run FlashWeave and EnDED (environmentally-driven-edge-detection and computing Jaccard index), is publicly available at GitHub (https://github.com/InaMariaDeutschmann/GlobalNetworkMalaspinaHotmix) and Zenodo[37].

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

## Acknowledgements
We thank all members of the Malaspina and Hotmix expeditions and the multiple projects funding these collaborative efforts. Sampling was carried out thanks to the Consolider-Ingenio program (project Malaspina 2010 Expedition, ref. CSD2008–00077, to C.M.D.) and HOTMIX project (CTM2011-30010/MAR, to J.A.), funded by the Spanish Ministry of Economy and Competitiveness Science and Innovation. Bioinformatic analyses have been performed at the Marbits computing core at the ICM-CSIC (https://marbits.icm.csic.es) and at the Supercomputing Center of Galicia (CESGA). I.M.D., R.L., and R.M. received funding from the European Union's Horizon 2020 research and innovation program under the Marie Skłodowska-Curie grant agreement no. 675752 (SINGEK, http://www.singek.eu). R.L. was supported by a Ramón y Cajal fellowship (RYC-2013-12554, MINECO, Spain). This work was also supported by the projects INTERACTOMICS (CTM2015-69936-P, MINECO, Spain), MicroEcoSystems (240904, RCN, Norway), and MINIME (PID2019-105775RB-I00, AEI, Spain) to R.L. S.C. was supported by the CNRS MITI through the interdisciplinary program Modélisation du Vivant (GOBITMAP grant). S.C., D.E., and S.G.A. were funded by the H2020 project AtlantECO (award number 862923). We acknowledge funding of the Spanish government through the "Severo Ochoa Centre of Excellence" accreditation to the ICM-CSIC (CEX2019-000928-S).

## Author contributions
The overall project was conceived and designed by R.L. J.M.G., C.M.D., S.G.A., R.M., and J.A. were responsible for the sampling and acquisition of contextual data. C.R.G., J.P., and M.S. processed specific samples in the laboratory. R.L. processed the amplicon data generating the two ASV tables. They were the starting point of the present study, which is part of the overall project. IMD developed the conceptual approach, and D.E., S.C., and R.L. contributed to its finalization. I.M.D. performed the data analysis. E.D., M.S., C.M.D., S.G.A., R.M., J.M.G., D.E., S.C., and R.L. contributed to interpreting the results. I.M.D. wrote the original draft. All authors contributed to manuscript revisions and approved the final version.

## Competing interests
The authors declare no competing interests.
