## [Peer Review File · Nature Communications]

Disentangling microbial networks across pelagic zones in the tropical and subtropical global oceanREVIEWER COMMENTS

Reviewer #1 (Remarks to the Author):

The authors of this study present a very complex network association analysis of datasets of the Malaspina expedition and a longitudinal transect across the Mediterranean Sea. They show that station- and region-specific network associations of the picoplankton communities including bacteria, archaea and eukaryotes, in the near surface ocean (3 m, deep chlorophyll maximum), the mesopelagic and bathypelagic zones exhibit differences and that there is a core microbiome in these oceanic regions. The level of the data analysis is rather abstract and does not go too much into details of the data and their interpretation. In the last eight years, several analyses have been published which assessed global oceanic diversity patterns of prokaryotic and eukaryotic communities, based on richness, diversity indices and various types of network associations. These analyses are based, predominantly but not exclusively, on samples collected during the Tara Ocean and Malaspina expeditions, and focussed on near surface, but including also deeper water layers down to the bathypelagic zone. They include quite a few publications of the authors consortium of this manuscript. After reading this manuscript the main question which remains and needs to be answered is what is really new and what kind of new and possibly unexpected findings are presented which really go beyond previous publications and support publication in this journal. In the epipelagic zone, such analyses have been published (Lima-Mendez et al. 2015, Chaffron et al. 2021, ref. 2 and 28 in this manuscript). Going beyond these analyses, there are certainly the detailed station- and region-specific network associations from the surface to the bathypelagic zone showing region-specific network-associations in particular in greater depth regions. However, it has already been shown that in particular in these depth regions the diversity of the picoplankton community and the prokaryotic community of different size fractions exhibit region- or ocean basin-specific features. These findings imply that there are also different network associations (Mestre et al. 2020, Villarino et al. 2022, refs. 11 and 45 in this manuscript) even though they have not been addressed specifically in these publications.

In addition to these rather general aspects there are more specific points which need to be addressed.

Interactions include taxonomic but mainly functional features. How would an association

network analysis cope with taxonomic and functional redundancies. Would they show up in such analysis or how would such an analysis be interpreted in the light of such redundant or non-redundant features?

The authors write at many places, including the title, that they carried out an analysis of a global data set. However, the Malaspina expedition was restricted to the subtropical and tropical global oceans, not going beyond 30° latitude, and the data of the Mediterranean Sea are also restricted to a latitudinal region extending not too much further north. It needs to be made clear that this is not an analysis of a global ocean data set.

At multiple places the authors write network associations and imply that they are biotic interactions. The rationale of applying network associations to infer biotic interactions is given in the Introduction and also used in many other publications. At few places the authors explicitly point out that network associations cannot be directly deduced from such associations and name them potential interactions. Throughout the manuscript there are only a few attempts made to identify and interpret these potential interactions in a meaningful way, content-wise, going beyond strong and weak interactions. Could they be similar in different regions and depth layers? Do they include more eukaryotes or bacteria or archaea? To distinguish "true" biotic interactions from environmentally driven co-occurrence patterns and to remove the latter they apply a pipeline they recently developed (EnDED). Its application to time series data was shown in this publication (Deutschmann et al. 2021, ref. 77). Is this pipeline really suitable to be applied to large-scale spatial data? Regarding the distinction of regional water masses it may be even suitable to include abiotic, e.g. hydrographic variables to characterize the specific features.

These basic points need to be addressed before a decision can be made whether the manuscript is suitable for publication in this journal.

In addition there are points which need to be specifically addressed.

I. 30: start the Abstract by saying first that you used association networks and deduce potential interactions.

I. 31: Be precise and mention that your analysis is restricted to the (sub)tropical oceans and Mediterranean Sea.

I. 38: There is no microbial sinking but sinking of PA microbes.

I. 39-40: Do you really think that such analyses can help conservation efforts to preserve the

ocean interactome? Must it be preserved under changing climate conditions or does it rather adapt?

I. 47: You may want to introduce here TINA as you showed in a previous publication how important these interactions are to respond to dynamics of environmental variables in the ocean.

I. 49: As written above this study is restricted to the tropical and subtropical oceans. A recent study expanded this approach beyond to temperate and subpolar regions and thus emphasized even more the role of temperature and other variables (Milke et al 2022). The latter study shows that the relative significance of homogenizing selection decreases when temperature differences of compared regions increases and that then dispersal limitation becomes relatively more important.

I. 59: better write be resolved

I. 71: even though this wording is catchy (and used by the TaraOcean consortium), it does not reflect the true range of stations of the Tara Ocean dataset. The southernmost stations were not even beyond the Antarctic Circle. Reword.

I. 75-77: Here you need to document the multiple studies as references.

I. 108-110: be precise and write that you used samples from the (sub)tropical regions of these oceans.

I. 119: It would be very helpful for a better understanding to provide an overview on the ASV data of bacteria, archaea and eukaryotes and on richness in each region and depth layer, absolute numbers and shared fractions of regions and depth layer, in a supplementary table and/or Venn diagram.

I. 121: You can be more precise and state that surface networks had a higher variability of nodes and edges than the other depth layers, also a trend to higher values.

I. 130-131: what is the benefit to compute subnetworks for each sample instead of each region? How robust are the station-specific networks?

Fig. 2 B: It does not really make sense to show prevalences of edges in sample-subsets of only 1 or a few samples! It seems like some region-depth layer-combinations are undersampled to make robust statements. The problem of low sample numbers in region-depth layer combinations affects further analyses: Fewest samples at DCM despite high numbers of associations compared to e.g. the meso- or bathypelagic. How does uneven sample number affect results shown in I 171-182?

L. 136-137: Why should highly prevalent interactions be crucial for ecosystem function: Consider the role of the Black Queen Hypothesis, in which prokaryotes of low abundance may exhibit crucial ecosystem services, like supply by a B vitamin or a growth hormone to phytoplankton algae.

I. 148-150: Give some more details on phylogenetic groups of the major classes here. The sentence not logic. Were Thaumarchaeota also present in the surface and Alphaproteobacteria also present in the deeper zones?

I. 177-179: SAO, SPA, IO had no decrease of associations with depth. why?

I. 198-200: awkward sentence: Can (potential) interactions be transferred or would the microbes be transferred such that interactions are formed newly or be modified including other residing microbes. Further, only a minor proportion of microbes sinks, that which is particle-associated. But your sampling design left out this fraction as potentially connecting link which weakens your argumentation.

L. 205-215: There are more comprehensive ways how to analyze the effect of external variables on interaction structure, such as TINA (using e.g. not SparCC but other inference methods that correct for indirect associations). You should elaborate these other options in the Discussion.

I. 246-248: Mention also that competition stabilizes interactions, associations. How would size- or group-specific grazing show up in your network associations?

I. 253-254: what would you consider as weak interactions? your statement further below is too simple.

L. 254-256: Discuss network stability in light of recent work, like Chaffron et al. 2021 (ref. 28).

I. 258-260: might this be a methods bias? Is the global network based on globally occurring ASVs?

Your "global" oceans are restricted to subtropical and tropical regions whereas Tara included regions beyond 30° latitude.

L. 263-281: The hypothesis of globally distributed ASVs is not robust in light of the last decade of marine microbiome research, at least for the epipelagic zone. Detection probability of an ASV is mostly linked to its environmental preferences (selection), less by dispersal limitation (see e.g. global connectivity by Villarino et al. 2018). Stochastic processes such as historical contingency play a larger role in higher depth and therefore

global ASVs can theoretically exist in deep environments that represent rather homogenous environmental conditions. The authors consequently discussed their findings more in light of biogeography of ASVs. Therefore, I wonder what is the new finding of this analysis?

I. 261-263: So methods seem to affect the outcome of the association networks, not only size fractions. Tara used OTUs and you ASVs with a much higher resolution. You should comment on this.

I. 263-265: In contrast to the previous comparison here you compare to Malaspina data whereas the difference are with Tara data. And you compare AVS and OTU data. Please clarify.

I. 268-269: this is expected as a paper by rather the same authors group reported of basin-specific picoplankton communities in the deep ocean (Villarino et al. 2022, ref. 45).

I. 273-275: is the complexity of ecosystems in the deep ocean really higher than in shallow depths? I guess that the opposite is the case. Everything is slowed down, Photoautotrophic phytoplankton is absent, grazing much reduced, growth rates and active interactions very low, due to the low numbers of microbes. Villarino et al 2022, argue that the deep ocean is more homogeneous than the epipelagic ocean as the prokaryotic communities in the deep ocean are more similar, in contrast to picoeukaryotes. What about this contradiction basically by the same lead authors?

I. 295-297: You mentioned before that geographic factors shape network topology to a lesser extent than environmental conditions, but here you highlight the effect of regional processes. What are the effects of regional vs. environmental processes? Can you discuss these points more in context of your data, also regarding the use of EnDED?

I. 310-314: Very long sentence. Split off.

I. 325—328: How can your methods detect changing interaction partners? According to your approach, local interactions are subsets from the global interactions, inferred based on the complete dataset. That means, when ASV A is present at the surface and at higher depth and has an interaction with ASV B that is only present at higher depth, then ASV A and ASV B do not express significant co-occurrences, right?

I. 334-337: This argument clearly shows the problem with co-occurrence analyses: The inferred network is strongly affected by methodology. Further, the ecological interpretation of network-indices such as average-path-length or network-connectivity is not straightforward. You conclude here that changes in the interactome along the water-column are

evident, but an ecological relevance of that statement is missing. Please elaborate further on that issue.

I. 358-361: Could you find more “endemic” interactions when you run co-occurrence network inference for each region/depth-layer individually? As you stated before, interactions can change, but this would be less detectable with a global network. You are welcome to discuss that point further.

I. 366-372: The conclusions are weak. What do your data really tell as new insights? What do you mean with “specific biogeographies which do not mirror taxonomic biogeographies”? What are you referring to here?

I. 394: must be assigned taxonomically.

I. 398-400: How do you account for a possible bias due to different size-fractions in network inference? Different size-fractions may have different (relative) abundances for the same taxa due to exclusion or accumulation by filter clogging of certain groups of organisms.

I. 412-413: How many ASVs were included in the analysis (after filtering)?

L. 424: Please elaborate on how EnDED works (how it identifies indirect edges). The number of indirect edges seems very low, especially that of temperature.

I. 466-467: What is the difference between point i) and ii) ? Also, please elaborate on the difference between Supplementary Figure 5A and B. Please also elaborate how many ASVs/Edges were filtered out using that procedure.

Reviewer #2 (Remarks to the Author):

Dear Authors and Editor,

Thank you for the opportunity to review the manuscript “Disentangling microbial networks across pelagic zones in the global ocean” by Deutchmann and colleagues, submitted to Nature Communications.

This manuscript makes the novel and important observation that the patterns of statistical associations between microorganisms generally differ between ocean regions and depths. In particular, they note that there are few globally consistent statistical associations, and few that are present across all depths – rather most associations are geographically local.

I think that this is an important point, and by itself makes this manuscript worthy for

publication in Nature Communications. Additionally this is, to my knowledge, the first comprehensive network analysis approach applied to Malaspina data (combined with the Hotmix expedition) which is also an important step.

However I had some challenges understanding much of their methods, as well as the the significance of some of their figures and points. I think the manuscript could be made more readable with consideration of my flowing comments:

Line 33: “that do not correspond to taxonomic distributions” What does it mean that heterogenous distributions do not respond to taxonomic distributions. I presume “distributions” mean something different in each case. In the former, its that interactions differ by ocean basin and depth. I’m not sure what “distributions” mean in the later case.

Line 38: What does “Despite microbial sinking” mean in this context. Are they detecting microbial sinking? Or do they contend that one would expect that since microbes are transported by marine snow, one would see the same interactions throughout the water column. If the later, I don’t see why microbial transport would be expected to lead to different inteeractions. Could “despite microbial sinking” be removed? Is so I think the sentence would be just as strong.

Line 39: I kind of like calling this the interactome. Great buzzword, makes network analysis seem like its on the -omics train. 100% in favor. However, I’m a little nervous that the authors aren’t really detecting interactions. They are actually detecting statistical correlations. Which might be related to interactions. Maybe some justification of this term, somewhere in the body of the paper is in order. Then I the rest of us networks people can start referring to our own networks as “interactome” studies, cite this one and get in on that -omics credibility.

In any case, they do a pretty good job of this on line 62, but should make the case for calling it an interactome there too. Or they could call these something other than an interactome (which I am not requesting).

Line 69: Somebody at some point should do an analysis like this one including the TARA oceans data in addition to this data set. Somewhere the authors should explain why they excluded this publicly available information.

Somewhere around line 74: the authors should cite Kara et al. 2012. This team broke spatial analysis of one location up by season and showed that the ... interactome differed between times of the year. Their own reference 33 does this as well, of course.

Kara EL, Hanson PC, Hu YH, Winslow L, McMahon KD. A decade of seasonal dynamics and co-occurrences within freshwater bacterioplankton communities from eutrophic Lake Mendota, WI, USA. ISME J. 2012 Oct 11;7:680–4.

Line 98: “Recent approaches we developed...” A short overview of how these approaches work would improve readability and not require all readers to find the other manuscript.

Line 114: Unless its a requirement of the journal, I think the paper might read better in the usual order with methods before results. The methods are novel, and kind of necessary to understand the results, I don't think the order is critical though, so the authors are welcome to disregard this suggestion.

Line 238: Misplaced “)”

Line 246: What sorts of future designs could detect weak interactions?

Line 274: Has it been established that the deep ocean system has “higher complexity”. If so, in what way?

Around Line 350: I encourage the authors to discuss how the patterns in the Mediterranean Sea might be representative of patterns in seas generally, as opposed to specific to the Mediterranean Sea. I realize they have not surveyed other seas and this is an oceans + med sea paper.

Line 389: Don't 515F-Y and 926R also get eukaryotes? How did they handle the overlapping eukaryotic data.

391. "in 5" is not grammatical.

Line 444. "Third, microorganisms needed to appear together (intersection) in more than 20% of the samples." Why? This seems like it would select against strong negative associations. One could imagine if two species just can't co-occur with each other for some reason, then this filter would make them not associated. Thus all of the negative associations must be weak and have exceptions to make it through the analysis. Perhaps this is why they have so many positive edges.

Line 501: "as proposed in 83. " is not grammatical as written. Add the Yaveroğlu et al. before the superscripted number.

Figure 2: How is the layout of the main network defined? How did the authors arrange the nodes so that the edges tend to cluster by region, or does that happen automatically? What is "association prevalence" as written, I don't understand the x-axis of the histograms.

Figure 3: I don't entirely follow what is happening in this figure. What are the nodes and edges exactly? They aren't described in the legend.

Figure 5: Requires clean up. The facet labels and text are all very small. There are many edges but they are impossible to distinguish from each other. Frankly, I'm not sure what I'm supposed to be taking away from this figure. Indeed, the figure caption, as in Figure 3, doesn't really explain what is going on, and the reader is left to guess.

Figure 6: Warrants more explanation. I could really use a plain language summary, or even a somewhat technical one of what a minimal spanning tree is, and why it is informative. I like

the idea of networks about networks and they look neat, but I don't really follow them.

Reference 33 is published. Please cite main paper.

We thank both reviewers for their time and all the comments and suggestions that helped to improve our work. Please find below our point-by-point answer to the reviewer's comments (in blue) and the actions taken, including changes in the main text. We also provide a version of the manuscript with tracked changes. Please note that the indicated lines of the changes refer to the document with tracked changes.

REVIEWER COMMENTS

Reviewer #1 (Remarks to the Author):

The authors of this study present a very complex network association analysis of datasets of the Malaspina expedition and a longitudinal transect across the Mediterranean Sea. They show that station- and region-specific network associations of the picoplankton communities including bacteria, archaea and eukaryotes, in the near surface ocean (3 m, deep chlorophyll maximum), the mesopelagic and bathypelagic zones exhibit differences and that there is a core microbiome in these oceanic regions. The level of the data analysis is rather abstract and does not go too much into details of the data and their interpretation. In the last eight years, several analyses have been published which assessed global oceanic diversity patterns of prokaryotic and eukaryotic communities, based on richness, diversity indices and various types of network associations. These analyses are based, predominantly but not exclusively, on samples collected during the Tara Ocean and Malaspina expeditions, and focussed on near surface, but including also deeper water layers down to the bathypelagic zone. They include quite a few publications of the authors consortium of this manuscript. After reading this manuscript the main question which remains and needs to be answered is what is really new and what kind of new and possibly unexpected findings are presented which really go beyond previous publications and support publication in this journal. In the epipelagic zone, such analyses have been published (Lima-Mendez et al. 2015, Chaffron et al. 2021, ref. 2 and 28 in this manuscript). Going beyond these analyses, there are certainly the detailed station- and region-specific network associations from the surface to the bathypelagic zone showing region-specific network-associations in particular in greater depth regions. However, it has already been shown that in particular in these depth regions the diversity of the picoplankton community and the prokaryotic community of different size fractions exhibit region- or ocean basin-specific features. These findings imply that there are also different network associations (Mestre et al. 2020, Villarino et al. 2022, refs. 11 and 45 in this manuscript) even though they have not been addressed specifically in these publications.

Answer: We thank the reviewer for this comment. Compared to the previous studies, we believe that our work makes the following novel contributions:

1. We analyze microbial association networks from the entire water column, particularly the deep ocean, across broad oceanic regions (tropical and subtropical), and the Mediterranean Sea. Previous work using association networks has concentrated on the

upper ocean layers (mostly epi-pelagic) ^{1,2}. Here, we leverage a unique dataset that covers the entire water column from the tropical and subtropical global ocean and the Mediterranean Sea, including several samples from the meso and bathypelagic layers.

2. We delineate sample-specific networks, which allow us to investigate how networks change over the vertical and horizontal dimensions of the ocean, across regions and depth layers.
3. Even though the turnover of the ocean microbiome has been analyzed in previous studies ³⁻⁸, the change of microbial association networks with depth across multiple locations of the ocean has seldom been investigated.

As mentioned by the reviewer and above, previous works have analyzed how the diversity of microbial communities changes with ocean depth. Still, they do not investigate how association networks change with depth over the entire water column across multiple locations or basins. We agree that changes in microbial diversity with depth imply a change in network structure. Yet, it is still necessary to analyze the association networks to determine their actual change and the emerging patterns, as changes in microbial diversity with depth could lead to multiple network topologies.

Action:

We have clarified the contribution of this work in the main text (Discussion):

L294: “Thus, compared to previous works that have focused on the surface layer^{27,29}, a critical, innovative aspect of this study is the analysis of the deep ocean interactome. Specifically, a key novel contribution of our work is analyzing how the ocean interactome changes with depth across the tropical and subtropical global ocean and the Mediterranean Sea. Even though previous works have already shown how microbial communities change with depth^{12,15,36,37}, our work goes further by showing how potential microbial interactions and derived network topologies change with depth.”

In addition to these rather general aspects there are more specific points which need to be addressed.

Interactions include taxonomic but mainly functional features. How would an association network analysis cope with taxonomic and functional redundancies. Would they show up in such analysis or how would such an analysis be interpreted in the light of such redundant or non-redundant features?

Answer: Different works using networks based on the 16S and 18S rRNA gene markers (e.g., ^{1,2,9,10}) make similar assumptions regarding the interpretation of the obtained signal in terms of potential interactions. Specifically, in networks based on these markers, we assume that the taxonomy may capture functional traits involved in the interactions through effects in microbial abundances (co-occurrences or co-exclusions). A specific microbial interaction will need to have enough specificity to be detected over space or time as a co-occurrence or co-exclusion pattern. However, promiscuous interactions due to functional redundancy (for example, one microbe that

can interact with several different ones that are ecologically equivalent) may not have enough abundance signal and may not be detected. This is a limitation of the approach, and in general, the limitations of the association networks approach have been discussed elsewhere ¹¹. In any case, we are confident that the signal we obtain using microbial abundances based on rRNA gene markers reflects strong spatial associations with substantial chances to represent ecological interactions. Even though its known limitations, this approach is still one of the best tools available to address the vast complexity of natural microbial interactions. Other approaches, such as metabolic modeling and metabolomics, are likely needed to determine the role of functional redundancy in microbial interaction networks. Yet, these approaches are starting to be implemented in environmental studies, and more development is needed to apply them to complex marine communities over large spatiotemporal scales.

Action: We have included the following text in the discussion, which is a modified version of the answer above:

L334: “In both our work and that of others using networks based on the 16S and 18S rRNA gene markers (e.g., ^{26,27,29,36}), it is assumed that the taxonomy may capture functional traits involved in the interactions through effects in microbial abundances (co-occurrences or co-exclusions). A specific microbial interaction must have enough specificity to be detected over space or time as a co-occurrence or co-exclusion pattern. However, promiscuous interactions due to functional redundancy (for example, one microbe that can interact with several different ones that are ecologically equivalent) may not have enough abundance signal and may not be detected. Despite this and other limitations of the association networks, as discussed in detail elsewhere ⁴⁵, the signal we obtain using microbial abundances based on rRNA gene markers can reflect strong spatial associations with substantial chances to represent ecological interactions. Thus, even though its known limitations, association networks are still one of the best tools available to address the vast complexity of natural microbial interactions. Other approaches, such as metabolic modeling and metabolomics ⁴⁶, are promising in detecting the role of functional redundancy in microbial interaction networks. Yet, these approaches are starting to be implemented in environmental studies, and more development is needed to apply them to complex marine communities over large spatiotemporal scales.”

The authors write at many places, including the title, that they carried out an analysis of a global data set. However, the Malaspina expedition was restricted to the subtropical and tropical global oceans, not going beyond 30° latitude, and the data of the Mediterranean Sea are also restricted to a latitudinal region extending not too much further north. It needs to make clear that this is not an analysis of a global ocean data set.

Answer: We agree with the reviewer. This terminology was intended to convey the broad scale of the study, but we understand it needs to be more accurate.

Action: We now mention in the main text “tropical and subtropical global ocean” as we have done in previous publications using Malaspina data ^{4,7,12}.

At multiple places the authors write network associations and imply that they are biotic interactions. The rationale of applying network associations to infer biotic interactions is given in the Introduction and also used in many other publications. At few places the authors explicitly point out that network associations cannot be directly deduced from such associations and name them potential interactions. Throughout the manuscript there are only little attempts made to identify and interpret these potential interactions in a meaningful way, content-wise, going beyond strong and weak interactions. Could they be similar in different regions and depth layers? Do they include more eukaryotes or bacteria or archaea?

Answers:

1) We have tried to be careful and, simultaneously, not too repetitive in indicating that association networks represent potential microbial interactions. As mentioned by the reviewer, this assumption is made in virtually all publications using association networks.

2) In **Lines 182 - 199**, we provide a general identification and interpretation of potential interactions (Please see below). Furthermore, in Figure 3, we investigate how the potential interactions change across regions and depths in terms of taxonomy. For example, we indicate that associations between specific groups become more prevalent as we move into the deep ocean. We have tried to mention the main patterns without going into much detail, as this may considerably increase the extent of the manuscript. Yet, we now refer to the interested reader to the main resulting tables deposited in the companion GitHub that include detailed information.

Actions:

1) We have double-checked throughout the manuscript that we refer to associations or potential interactions, not just interactions when referring to association networks.

2) We now mention in the main text, **Line 219**, "Above, we have only described main patterns; see further details in the GitHub repository in Data Availability, sections 04_Prevalence, 05_ClassifyingAssociations, and 06_VerticalConnectivity"

To distinguish "true" biotic interactions from environmentally driven co-occurrence patterns and to remove the latter they apply a pipeline they recently developed (EnDED). Its application to time series data was shown in this publication (Deutschmann et al. 2021, ref. 77). Is this pipeline really suitable to be applied to large-scale spatial data?

Regarding the distinction or regional water masses it may be even suitable to include abiotic, e.g. hydrographic variables to characterize the specific features.

Answers:

1) EnDED was designed with microbial networks in mind and is suitable for large-scale spatial data ¹³. EnDED is an implementation of 4 methods and their combination. Some methods we implemented in EnDED were already used in a large-scale dataset from TARA Oceans ² to identify indirect edges. EnDED has been tested on simulated data, in particular time series, as one of the methods included leverages temporal data. But, EnDED has not been designed only for temporal data and can handle spatial datasets.

2) The environmental metadata analyzed considers variables correlated with water masses, such as salinity and temperature. These variables should contribute to the detection of indirect edges linked to different water masses.

Action: We have added to the main text, **Line 704**, “Thus, we applied EnDED⁷⁸, a method suitable for large-scale spatial data.”

These basic points need to be addressed before a decision can be made whether the manuscript is suitable for publication in this journal.

In addition there are points which need to be specifically addressed.

I. 30: start the Abstract by saying first that you used association networks and deduce potential interactions.

Answer & Action: Done. We now mention, **Line 30**, “Here, we use association networks to investigate possible ecological interactions in the marine microbiome among archaea, bacteria, and picoeukaryotes...”

I. 31: Be precise and mention that your analysis is restricted to the (sub)tropical oceans and Mediterranean Sea.

Action: Changed

I. 38: There is no microbial sinking but sinking of PA microbes.

Answer & Action: We have changed this. We now mention, **Line 38**, “ despite microbial vertical dispersal.”

I. 39-40: Do you really think that such analyses can help conservation efforts to preserve the ocean interactome? Must it be preserved under changing climate conditions or does it rather adapt?

Answer & Action: We have removed this sentence. Nevertheless, a better understanding of the ocean interactome could inform us on whether it is reasonable to implement conservation measures to, for example, prevent losing key interactions or if we should expect the interactome to adapt to global change. At the moment, knowledge is limited to tell us the best strategy. We agree it is a very interesting subject, beyond the scope of this paper, though.

I. 47: You may want to introduce here TINA as you showed in a previous publication how important these interactions are to respond to dynamics of environmental variables in the ocean.

Answer & Action: Thank you for this suggestion. We now mention, **Line 61**, “In addition, we previously found that potential interactions seem relevant in structuring the surface ocean microbiota”

I. 49: As written above this study is restricted to the tropical and subtropical oceans. A recent study expanded this approach beyond to temperate and subpolar regions and thus emphasized even more the role of temperature and other variables (Milke et al 2022). The latter study shows that the relative significance of homogenizing selection decreases when temperature differences of compared regions increases and that then dispersal limitation becomes relatively more important.

Answer & Action: Thanks for pointing this out. Now we cite: Milke, F., Wagner-Doebler, I., Wienhausen, G. & Simon, M. Selection, drift and community interactions shape microbial biogeographic patterns in the Pacific Ocean. ISME J 16, 2653–2665 (2022).

I. 59: better write be resolved

Action: Modified

I. 71: even though this wording is catchy (and used by the TaraOcean consortium), it does not reflect the true range of stations of the Tara Ocean dataset. The southernmost stations were not even beyond the Antarctic Circle. Reword.

Action: Modified

I. 75-77: Here you need to document the multiple studies as references.

References included

I. 108-110: be precise and write that you used samples from the (sub)tropical regions of these oceans.

Changed, now we mention, **Line 129** "...and tropical and subtropical areas of five ocean basins"

I. 119: It would be very helpful for a better understanding to provide an overview on the ASV data of bacteria, archaea and eukaryotes and on richness in each region and depth layer, absolute numbers and shared fractions of regions and depth layer, in a supplementary table and/or Venn diagram.

Answer: We thank the reviewer for this comment. Part of this information is given in "Supplementary Figure 4: ASVs across depth layers. For each region, we color ASVs based on the layer they first appeared: Surface [SRF] (S, yellow), DCM (D, orange), Mesopelagic [MES] (M, red), and Bathypelagic [BAT] (B, black). Next to each layer, the number of ASVs that were first detected in that layer is indicated. Absent ASVs are grouped in box "a". An ASV, only appearing in the bathypelagic, is assigned to box "a" in the above layers. That is, an ASV detected in the surface and present in the DCM but absent in lower layers appears in box (S) in the surface and DCM layer but in box "a" in the meso- and bathypelagic layer. An ASV cannot be assigned to two layers. Note that most ASVs in the bathypelagic zone have already been detected in upper layers because most ASVs are assigned to the boxes "S", "D", and "M" instead of "B". See specific details in the GitHub repository [section 06_Vertical Connectivity, Additional Tables]."

Action: We now include a new Supplementary Table 1 (please see below), where we indicate the number of ASVs in the layer where they were first detected and the number of ASVs that are unique to a depth layer. We also added citations to Supplementary Figure 4 in the main text. The number of Bacteria, Archaea, and Eukaryotes per depth layer, indicating those that are shared, is now provided in the companion GitHub repository (https://github.com/InaMariaDeutschmann/GlobalNetworkMalaspinaHotmix/tree/main/06_VerticalConnectivity/SupplementaryFigure4_AdditionalTables), given the large size of these tables. In addition, richness is now included in the new version Supplementary Figure 4 for each region and depth layer.

Line 140: "We found vertical and horizontal differences in the distributions of ASVs and the number of unique ASVs per depth layer (Supplementary Table 1, Supplementary Figure 4; see specific details in the GitHub repository [section 06_VerticalConnectivity, Additional Tables]), which is consistent with previous works ^{5,12,37-39}. Contrary to communities, we have a limited

understanding of how much marine microbial networks change due to dispersal as well as vertical and horizontal environmental heterogeneity. ”

Line 1219:

“**Supplementary Table 1:** Number of ASVs in the layer where they were first detected (from surface to bottom) for the different regions and, in brackets, ASVs unique to specific depth layers. Note that ASVs from the upper layers can be present in the lower layers, but not vice versa. For example, the 1358 ASVs first detected in the Mesopelagic of the Mediterranean Sea (MES) can be present in the Bathypelagic but not in the DCM. In turn, the ASVs that are unique to specific depth layers (in brackets) are only present in the specified layer.

	MS	IO	NAO	SAO	NPO	SPO
Surface	1116 (103)	2838 (2005)	3093 (1821)	2718 (1807)	2767 (1809)	2405 (1870)
DCM	635 (95)	399 (212)	189 (52)	491 (239)	289 (109)	199 (112)
Mesopelagic	1358 (326)	1198 (632)	1423 (227)	1113 (464)	1183 (547)	775 (431)
Bathypelagic	88 (88)	423 (423)	507 (507)	422 (422)	473 (473)	590 (590)

Line 1196: “Surface [SRF] (S, yellow), DCM (D, orange), Mesopelagic [MES] (M, red), and Bathypelagic [BAT] (B, black). Next to each layer, the number of ASVs that were first detected in that layer is indicated.”

Line 1203: See specific details in the GitHub repository [section 06_VerticalConnectivity, Additional Tables].

I. 121: You can be more precise and state that surface networks had a higher variability of nodes and edges than the other depth layers, also a trend to higher values.

Answer & Action: Done. **L149.** We now mention “..with surface networks tending to display higher values, and a higher variability, in the number of nodes and edges in the ocean basins”

I. 130-131: what is the benefit to compute subnetworks for each sample instead of each region? How robust are the station-specific networks?

Answer: Determining a subnetwork for each sample allows us to detect changes at smaller spatial scales than if we investigate networks in regions, which are spatially much larger. Also, if we would use entire regions, we would assume that all associations and ASVs are present within a given region in contrast to our findings showing that this is not the case. The previous is included in our third condition to build networks, which is mentioned in Methods:

“*iii. the association has a region and depth-specific Jaccard index, J_{RL} , above 20% .*

Regarding the third condition, we determined J_{RL} for each association pair by computing within each region and depth layer, the fraction of samples two microorganisms appeared together (intersection) from the total samples at least one microorganism appears (union). ”

Since station-specific networks derive from a global network, which was calculated including all samples, they are expected to be equally robust.

(NB: we have split the reviewer's question below to try to provide better answers)

Fig. 2 B: It does not really make sense to show prevalences of edges in sample-subsets of only 1 or a few samples!

Answer and Action: Thanks for indicating this. We have modified Figure 2B and removed the x-y subaxis for the DCM-SPO, for which there is only one sample (no prevalence is indicated now). We also mention in the figure “NB: one-sample network”.

It seems like some region-depth layer-combinations are undersampled to make robust statements. The problem of low sample numbers in region-depth layer combinations affects further analyses: Fewest samples at DCM despite high numbers of associations compared to e.g. the meso- or bathypelagic.

Answer: Thank you for raising this point. Unequal sampling efforts in regions and depth layers could have introduced biases, but this is complicated to control in large, global-ocean-scale, oceanographic cruises that include very complex logistics and multiple sampling teams. Despite these limitations, we were able to compile one of the largest datasets for the marine microbiome that includes different depth layers, and we are convinced our results can lead to important insights (please see our action point below).

How does uneven sample number affect results shown in I 171-182?

Answer: Unequal sampling efforts may have an impact on the regional associations for regions with only one or a few samples. In such cases, a larger proportion of edges may be considered, while in cases with more samples, potentially more edges could be filtered out. Unequal sampling efforts could also impact the identification of global associations. Some global associations may be missed in regions with few samples. In turn, other putatively global associations may be overestimated in cases where there is only one sample in a region, as in this case, all edges from the region are considered. Despite these potential biases, we aimed to focus on the strongest and broadest emerging patterns. We consider these to be sufficiently robust.

Action:

We have added to Discussion **L605** “Even though our dataset is one of the largest generated so far for the tropical and subtropical global ocean and the Mediterranean Sea, it includes different sampling efforts for regions and depth layers. This may have introduced biases in the detected

regional or global associations. Furthermore, different methodologies should be tested, as network construction tools, analytical thresholds, sequencing depth, and sampling design could all influence the amount of putatively endemic interactions that are detected. Despite these limitations, we consider that our work provides important insights into the amount of regional and cosmopolitan putative interactions in the tropical and subtropical global ocean and the Mediterranean Sea.”

L. 136-137: Why should highly prevalent interactions be crucial for ecosystem function: Consider the role of the Black Queen Hypothesis, in which prokaryotes of low abundance may exhibit crucial ecosystem services, like supply by a B vitamin or a growth hormone to phytoplankton algae.

Answer: Widespread microbial interactions are linked to equally widespread microbes that may have important roles in the ocean's cycling of nutrients and energy. As shown in other works, including our own ⁴, widespread microorganisms in the ocean tend to display higher abundances than organisms with lower abundances. Therefore they may have a higher impact on ecosystem processes (e.g., carbon fixation). For example, we could think of an abundant and widespread marine photosynthetic microbe that has a relevant influence on the carbon cycle, and that has a strong mutualistic interaction with another heterotrophic microbial species. Then, disrupting this mutualistic interaction could negatively impact the role of the photosynthetic microbe in the carbon cycle. In any case, we have also observed highly prevalent associations linked to low-abundance ASVs, which could also be linked to important ecosystem processes. We have changed the text in the manuscript to reflect this.

Action: We now mention in **L169**: “Highly prevalent associations present across all regions and linked to high or low abundance ASVs are candidates to represent putative core interactions in the tropical and subtropical global ocean that may be connected to processes that are important for ecosystem function.”

I. 148-150: Give some more details on phylogenetic groups of the major classes here. The sentence not logic. Were Thaumarchaeota also present in the surface and Alphaproteobacteria also present in the deeper zones?

Answer: Broad patterns of low-frequency, prevalent, and global associations at the phyla level are given in: “Supplementary Figure 2: Associations occurring in each region and depth layer and their taxonomy. If an association appears in more than 20% of subnetworks in each region, it is classified as low-frequency, >50% prevalent, and >70% global. The number of samples appear in the upper left corner, the number of edges in the upper right corner, and the depth range in the lower right corner (in meters [m] below the surface). We classified the associations considering all six regions A-D) and considering the five ocean basins without the MS, E-H).”

Additional details are provided at the companion GitHub:

https://github.com/InaMariaDeutschmann/GlobalNetworkMalaspinaHotmix/blob/main/04_Prevalence/HighlyPrevalentEdges_FrequencyPercentagePerRegionDepth.tsv

The following tables provide the complete list of edges given by pairs of ASVs and all classifications:

1. https://github.com/InaMariaDeutschmann/GlobalNetworkMalaspinaHotmix/blob/main/05_ClassifyingAssociations/AssociationClassification_AbsentPresent.tsv
2. https://github.com/InaMariaDeutschmann/GlobalNetworkMalaspinaHotmix/blob/main/05_ClassifyingAssociations/AssociationClassification_AbsentPresent_noMS.tsv

This folder contains the taxonomic information:

3. https://github.com/InaMariaDeutschmann/GlobalNetworkMalaspinaHotmix/tree/main/00_Tables (Euk_taxdata.txt and Pro_taxdata.txt)

Mapping Tax info as needed from 3. to the upper tables 1 & 2, and then grouping will give the desired info for any category. As there are many combinations that can be generated, we prefer to provide the data for the interested reader so they can explore the multiple possible combinations.

Note that the previous tables are very large, and therefore, we prefer to provide them in the companion GitHub repository.

Action:

We have modified this section in the main text, including more details. We prefer not to mention in the text the Class taxonomic level, as this would require substantial explanations and may not add to the main message of these results. For the interested reader, we have prepared the companion tables, available in GitHub (05_ClassifyingAssociations; see above), where all edges are listed via their ASVs and their classifications (for all regions and also when the MS is not considered). These tables, together with the taxonomic information, also available on GitHub, can be used by interested readers to dive deeper into microbial interaction hypotheses.

We now mention **L182**: “While Alphaproteobacteria were present in associations across depth layers, they dominated in the epipelagic global associations (Supplementary Figure 2). Still, Alphaproteobacteria were well represented in global associations in the bathypelagic (Supplementary Figure 2). Dinoflagellates were also well represented in epipelagic global associations. Most global associations from the mesopelagic and bathypelagic included Thaumarchaeota, which were more abundant in deeper zones (Supplementary Figure 2). Dinoflagellates and Alphaproteobacteria tended to be common among epipelagic Prevalent and Low-Frequency associations, while other taxonomic groups displayed a more variable

representation (Supplementary Figure 2). In the mesopelagic and bathypelagic, Thaumarchaeota and Alphaproteobacteria tended to be common among Prevalent and Low-Frequency associations, yet other lineages were prevalent in specific cases, such as the eukaryotic SAR, Dinoflagellates, Radiolarians, Actinobacteria, Deltaproteobacteria, and Gammaproteobacteria (See more details in Supplementary Figure 2 and in the GitHub repository [05_ClassifyingAssociations; here all edges are listed via their ASVs and their classifications]).”

I. 177-179: SAO, SPA, IO had no decrease of associations with depth. Why?

Answer: In Figure 3, when we compare the epipelagic (Surface and DCM), with both the Mesopelagic and Bathypelagic, we observe a decrease in the number of associations with depth for the SAO, SPO, and IO.

I. 198-200: awkward sentence: Can (potential) interactions be transferred or would the microbes be transferred such that interactions are formed newly or be modified including other residing microbes. Further, only a minor proportion of microbes sinks, that which is particle-associated. But your sampling design left out this fraction as potentially connecting link which weakens your argumentation.

Answer: Thanks for pointing this out. We agree this sentence needs to be clarified. We want to indicate that even though a substantial fraction of surface ASVs are also present in deep waters, their associations are not. This could be exemplified by two ASVs that are associated in surface waters but, if present in the deep ocean, they are not.

The presence of a substantial proportion of surface ASV in the deep sea has been reported in previous work, and this also includes the “free-living” fraction. Please see Figure 2 at <https://doi.org/10.1073/pnas.1802470115>.

Action: We have modified the two sentences below (**Line 251**). Please note that both sentences should be interpreted together:

“In addition, most surface associations disappeared with depth in the five ocean basins and MS (Figure 5), suggesting that most surface ocean interactions among the picoplankton are not transferred to the deep sea, despite the vertical dispersal of various taxa¹². Specifically, we observed that most deep ocean ASVs already appeared in the upper layers (Supplementary Figure 4), in agreement with previous work that has shown that a large proportion of deep-sea microbial taxa, including those from small size-fractions, are also found in surface waters, and their presence in the deep sea is putatively related to sinking particles¹².”

L. 205-215: There are more comprehensive ways how to analyze the effect of external variables on interaction structure, such as TINA (using e.g. not SparCC but other inference methods that correct for indirect associations). You should elaborate these other options in the Discussion.

Answer: We agree that TINA (based on Flashweave, for example) would be an alternative for general analyses. However, it may not provide the resolution level provided by the method we have used, as TINA is typically implemented with “global” or regional networks. In addition, our approach gives substantial weight to the interconnection patterns of the network (graphlets), while TINA aims to quantify the similarity between two communities as the average interaction strength between all taxa observed in them ¹⁴.

Action: We now mention in the discussion:

Line 419 “We sought to understand the impact of environmental gradients and geographic factors on the topology (that is, patterns of network connections) of microbial networks in the tropical and subtropical global ocean across depths.”

Line 449 “These findings align with earlier research on global epipelagic networks ^{27,29} and a previous work where we found that prokaryotes inhabiting locations in the tropical and subtropical surface ocean featuring similar temperatures tend to co-occur ⁵. In the last study, we used the TINA index ⁴⁸, which aims to quantify the similarity between two communities as the average interaction strength between all taxa observed in them, while here, we increase the resolution of the environmental analyses by comparing the actual topologies of networks using graphlets. Altogether, and regardless of the analysis type, multiple pieces of evidence point to environmental heterogeneity having a substantial effect in shaping the topology of association networks in the ocean.”

I. 246-248: Mention also that competition stabilizes interactions, associations. How would size- or group-specific grazing show up in your network associations?

Answer: Thanks for indicating us to include competition as a stabilizing factor in interaction networks.

In the current manuscript, we have not explicitly investigated how size- or group-specific grazing would appear in our networks. Yet, we would expect non-selective grazing to be reflected in our networks as protistan ASVs or groups of protistan ASVs associated with multiple prokaryotes, as we have shown for the heterotrophic flagellate MAST-4 ¹² and also in agreement with findings reported by other authors ¹⁵.

Action: We now mention, **Line 320** “Weak interactions and competition (or other negative interactions) are essential for community stability, as networks including these interactions are less prone to destabilizing cascade effects ^{38–43}”. The following reference was added ¹⁶

I. 253-254: what would you consider as weak interactions? your statement further below is too simple.

Answer: weak interactions are defined here as those associations between microbial taxa that exhibit subtle or low-magnitude correlations.

Action: We have modified the section below in Discussion to address this point:

Line 316 “ Alternatively, it is also possible that the sampling design and methodological approach missed the majority of negative, or weak interactions (defined here as those associations between microbial taxa that exhibit subtle or low-magnitude correlations) ²⁶. For example, plummeting species abundances between stations or samples could prevent establishing significant negative correlations. Weak interactions and competition (or other negative interactions) are essential for community stability, as networks including these interactions are less prone to destabilizing cascade effects ^{37–42}. Considering that the horizontal turnover of the abundant taxa populating the ocean microbiome is usually gradual rather than drastic ^{5,8}, stabilizing mechanisms such as weak interactions and competition are expected to be common. In addition, recent work has shown that networks' vulnerability to global change can differ across ocean regions, being particularly high in the Arctic ²⁹. Therefore, investigating interactions that stabilize networks or make them more resilient to disturbances is of particular interest, and future work should implement sampling designs and analytical approaches that can better characterize them. For example, high-frequency sampling campaigns including multiple replicates, microcosms and manipulative field experiments, and metabolic modeling coupled to metabolomics⁴⁵.”

L. 254-256: Discuss network stability in light of recent work, like Chaffron et al. 2021 (ref. 28).

Action: We have now added, **Line 327** “ In addition, recent work has shown that networks' vulnerability to global change can differ across ocean regions, being particularly high in the Arctic ²⁹. Therefore, investigating interactions that stabilize networks or make them more resilient to disturbances is of particular interest, and future work should implement sampling designs and analytical approaches that can better characterize them.”

I. 258-260: might this be a methods bias? Is the global network based on globally occurring ASVs?

Your “global” oceans are restricted to subtropical and tropical regions whereas Tara included regions beyond 30° latitude.

Answers:

1) We do not think it is a method bias, as the work of Chaffron et al., ¹ shows similar results. We think that the main reason for the differences with the results of Lima-Mendez et al. ² is based on the methodology. Our work and that of Chaffron and colleagues use comparable methods that differ from those used by Lima-Mendez and colleagues. This is now indicated in the new version of the paragraph (please, see below).

2) The global network does not include globally occurring ASVs only. Our definition of global associations is:

“We considered an association to be global within a specific depth layer if its prevalence was above 70% in all regions”

Therefore, OTUs being part of global associations for a given depth layer should have a prevalence above 70% in all regions.

We have clarified now in the text what we mean by global.

Action: We have modified the paragraph below to address the points mentioned above. Now it reads (**Line 349**):

“By using an innovative approach to determine sample-specific interactomes, we identified global (i.e., present in all regions with >70% prevalence for a given depth layer) and regional pelagic microbial associations across the oceans’ vertical and horizontal dimensions. We found few global associations, indicating a potentially small core interactome in the tropical and subtropical global ocean and the Mediterranean Sea within each depth layer. Our results agree with those from Chaffron et al. ²⁹, which showed that global or widespread associations were a minority and that most associations were community-specific. These results align with those reporting a relatively small number (between 0.3-1%) of widespread microbial taxa in the surface ocean ⁵, given that more geographically ubiquitous taxa could lead to more widespread interactions. In turn, these results contrast with those of Lima-Mendez et al. ²⁷, who found a large proportion of widespread associations in the global sunlit ocean. Most likely, this reflects differences in the used approaches and datasets, as our work and that of Chaffron et al. ²⁹ implement a recently developed network construction tool that infers direct associations from heterogeneous microbial abundance datasets (FlashWeave ⁴⁹), and, in addition, both works analyzed sample-specific networks. The previous analyses were not implemented by Lima-Mendez and colleagues ²⁷, and this could be a reason for the observed differences. Both mentioned studies were based on TARA Oceans datasets and have used OTUs or miTags ⁵⁰, while here, we have used ASVs with higher taxonomic resolution ⁵¹, which could also contribute to differences between the results. ”

L. 263-281: The hypothesis of globally distributed ASVs is not robust in light of the last decade of marine microbiome research, at least for the epipelagic zone. Detection probability of an ASV is mostly linked to its environmental preferences (selection), less by dispersal limitation (see e.g. global connectivity by Villarino et al. 2018). Stochastic processes such as historical contingency play a larger role in higher depth and therefore global ASVs can theoretically exist in deep environments that represent rather homogenous environmental conditions. The authors consequently discussed their findings more in light of biogeography of ASVs. Therefore, I wonder what is the new finding of this analysis?

Answer and action: We now use the words “widespread” and “ubiquitous” rather than “cosmopolitan”, which better reflects our message. Specifically, we refer to ASVs that could be in, for example, >80% of the samples, as we did in previous work⁴. The modified sentence reads:

Line 362 “These results align with those reporting a relatively small number (between 0.3-1%) of widespread microbial taxa in the surface ocean⁵, given that more geographically ubiquitous taxa could lead to more widespread interactions.”

In the new version of the manuscript, we tried to clarify and focus the discussion on the distributions of associations and not on that of ASVs.

I. 261-263: So methods seem to affect the outcome of the association networks, not only size fractions. Tara used OTUs and you ASVs with a much higher resolution. You should comment on this.

Answer and action: Thanks for this suggestion, we have tried to clarify this section. We now also mention the differences between OTUs and ASVs:

Line 364 “In turn, these results contrast with those of Lima-Mendez et al.²⁷, who found a large proportion of widespread associations in the global sunlit ocean. Most likely, this reflects differences in the used approaches and datasets, as our work and that of Chaffron et al.²⁹ implement a recently developed network construction tool that infers direct associations from heterogeneous microbial abundance datasets (FlashWeave⁴⁹), and, in addition, both works analyzed sample-specific networks. The previous analyses were not implemented by Lima-Mendez and colleagues²⁷, and this could be a reason for the observed differences. Both mentioned studies were based on TARA Oceans datasets and have used OTUs or miTags⁵⁰, while here, we have used ASVs with higher taxonomic resolution⁵¹, which could also contribute to differences between the results.”

I. 263-265: In contrast to the previous comparison here you compare to Malaspina data whereas the difference are with Tara data. And you compare AVS and OTU data. Please clarify.

Answer and action: This sentence was moved up to improve clarity (**Line 362**)

I. 268-269: this is expected as a paper by rather the same authors group reported of basin-specific picoplankton communities in the deep ocean (Villarino et al. 2022, ref. 45).

Answer and action: We have tried to clarify this section by providing additional explanations and incorporating more references. Now the text reads:

Line 380 "This may reflect a higher dispersal limitation in deep ocean regions due to slow currents, water masses⁴⁷, straits, and seamounts⁴⁸. Consistently, using the same picoplankton dataset, we observed that selection tends to decrease while dispersal limitation tends to increase with depth in the tropical and subtropical global ocean and the Mediterranean Sea⁴⁹. Other studies have also reported a higher dispersal limitation with depth in specific ocean regions³¹. A recent study found increasing differences in picoplankton community composition with depth when comparing the surface vs. the deep layer of different ocean basins³⁷. Yet, when analyzing the entire dataset encompassing the tropical and subtropical ocean, the previous work found that surface prokaryotic communities were more different across stations than deep counterparts, while surface picoeukaryotic communities were more similar than deep ones. This could be attributed to the differential action of selection and dispersal limitation in prokaryotes and picoeukaryotes across water layers^{37,49}."

I. 273-275: is the complexity of ecosystems in the deep ocean really higher than in shallow depths? I guess that the opposite is the case. Everything is slowed down, Photoautotrophic phytoplankton is absent, grazing much reduced, growth rates and active interactions very low, due to the low numbers of microbes. Villarino et al 2022, argue that the deep ocean is more homogeneous than the epipelagic ocean as the prokaryotic communities in the deep ocean are more similar, in contrast to picoeukaryotes. What about this contradiction basically by the same lead authors?

Answer and action: We thank the reviewer for pointing out this inconsistency, which was based on our lack of clarity. We have now tried to clarify this section that reads:

Line 411 "Even though environmental gradients in the deep ocean are smooth when compared to the surface⁴⁷, specific processes could contribute to increasing the complexity and number of niches in the deep ocean, potentially leading to an increase in the number of regional associations. For example, different niches may be associated with the quality and types (labile vs. recalcitrant) of organic matter reaching the deep ocean from the epipelagic zone³⁰, which is significantly different across oceanic regions⁵⁰. In an exploration of generalists versus specialist prokaryotic metagenome-assembled genomes (MAGs) in the Arctic Ocean, most specialists were linked to mesopelagic samples, indicating that their distribution was uneven across depth layers⁵¹."

We have also added to the beginning of the previous paragraph:

Line 374 "Ocean currents are generally stronger, and the variability in environmental conditions is greater in surface waters compared to the deep ocean⁴⁷. This could affect the relative importance of selection and dispersal in structuring microbial communities and, consequently, association networks."

I. 295-297: You mentioned before that geographic factors shape network topology to a lesser extent than environmental conditions, but here you highlight the effect of regional processes.

What are the effects of regional vs. environmental processes? Can you discuss these points more in context of your data, also regarding the use of EnDED?

Answer and action: We have elaborated more on the regional/spatial vs. environmental processes in the text:

Line 463 “In a study of the surface global-ocean interactome Chaffron and colleagues²⁹ found few direct associations between taxa and environmental variables, similar to our findings when investigating a marine-coastal interactome over ten years²⁶. For interactomes where positive associations predominate, such as the two previous works plus others^{24,27,56}, this suggests that environmental variables may significantly influence a number of species. These would then pull the others, a process facilitated by positive associations, generating cascade effects and specific dynamics³⁸, as we have previously suggested for a marine coastal interactome²⁶. It is worth reminding that we have removed indirect edges that reflect similar environmental preferences and not potential interactions by using FlashWeave coupled with EnDED (see Methods).

The observed differences in network topology across distinct but environmentally similar oceanic regions or basins suggest that regional processes also play a role in determining network topology. Different dispersal rates between ocean regions may be responsible for a substantial fraction of the regional effects. This aligns with recent results that found compelling evidence that ocean currents exert a significant basin-scale influence on microbial plankton biogeography⁵⁷. Furthermore, stochastic changes in community composition, or drift, could also underpin regional effects. Recent studies found evidence that dispersal, drift, and selection change with depth, among ocean basins, and between basins and the Mediterranean Sea^{5,9,49}. This variability in the relative importance of the ecological process shaping communities may explain, to a certain extent, the changes in network topologies across regions. Other unmeasured processes could also play a role, such as the promiscuity of interactions (that is, the possibility of one microbe establishing ecological interactions with the same or different partners across ocean regions⁵⁵).”

I. 310-314: Very long sentence. Split off.

Answer and Action: Done, this sentence now reads

Line 503 “While ASVs belonging to these taxonomic groups were present throughout the water column, specific associations were observed mainly in the mesopelagic and bathypelagic zones. This suggests specific interactions between endemic deep-sea taxa, in agreement with the hypothesis indicating high niche partitioning and more specialist taxa in the deep ocean^{59,60}.”

I. 325—328: How can your methods detect changing interaction partners? According to your approach, local interactions are subsets from the global interactions, inferred based on the complete dataset. That means, when ASV A is present at the surface and at higher depth and has an interaction with ASV B that is only present at higher depth, then ASV A and ASV B do not express significant co-occurrences, right?

Answer: The answer may be slightly different. We derive sample-specific networks from a general network where all samples are considered (please see details below). In the example provided, we think there could be an edge between ASV A and B in the general network due to co-occurrences in specific deep-sea samples. Yet, in sample-specific networks, ASV A and B will be present and connected in higher-depth networks only as per the approach below.

Subnetworks are inferred from the main network following this procedure (extracted from Methods):

“First, we require that an edge must be present in the single static network. Second, an edge can only be present within a subnetwork if both microorganisms associated with the edge have a sequence abundance above zero in the corresponding sample. Third, microorganisms associated need to appear together (intersection) in more than 20% of the samples, in which one or both appear (union) for a specific region and depth.

Formally, consider sample s_{RL} with R being the marine region and L the sample’s depth layer. Let e be an association between microorganisms A and B . Then, association e is present in the sample-specific subnetwork N_s , if

- I. e is an association in the single static network,
- II. the microorganisms A and B are present within sample s , i.e., the abundances are above zero within that particular sample, and
- III. the association has a region and depth-specific Jaccard index, J_{RL} , above 20% (see below).

In addition to these three conditions, a node is present in a sample-specific subnetwork when connected to at least one edge, i.e., we removed isolated nodes.”

One type of artificial association arises from structural zeroes like the one described by the reviewer, with a non-random absence of one association partner. Structural zeroes can also occur when a dataset includes multiple sequencing protocols. Due to our heterogeneous data compilation, we chose FlashWeave¹⁷, which aims to be robust to such absences. From a technical perspective, zeros are excluded, and therefore, if A and B co-occur significantly in the deep sea, FlashWeave should be able to detect that, irrelevant to the surface samples where only A but not B appear.

NB: we mention in Methods (**Line 690**): “We excluded rare microorganisms by keeping ASVs with a sequence abundance sum above 100 reads across all samples and appearing in at least 20 samples (>5% of the dataset). The latter condition removed eukaryotes only appearing in the 13 MalaDeep eukaryotic samples of the 0.8-20 μm size fraction. This filtering step left 2922 eukaryotic ASVs, representing 79.8% of the original eukaryotic reads, and 2535 prokaryotic ASVs, representing 84.8% of the original prokaryotic reads. To control for data compositionality⁹⁰, we applied a centered-log-ratio transformation separately to the prokaryotic and eukaryotic tables before merging them.

Second, we inferred a (preliminary) network using FlashWeave⁴⁹, based on 5457 ASVs selecting the options “heterogeneous” and “sensitive”. FlashWeave was chosen as it can handle sparse datasets like ours, taking zeros into account and avoiding spurious correlations between ASVs that share many zeros.”

I. 334-337: This argument clearly shows the problem with co-occurrence analyses: The inferred network is strongly affected by methodology. Further, the ecological interpretation of network-indices such as average-path-length or network-connectivity is not straight-forward. You conclude here that changes in the interactome along the water-column are evident, but an ecological relevance of that statement is missing. Please elaborate further on that issue.

Answer: We thank the reviewer for this comment. Given that we are comparing our results with those from other works that followed other methods (including sampling and analysis of sequences), it is expected that the methodology affects the results. For our results to be completely comparable, we should have followed the same methodology. Nevertheless, there are main patterns that emerge, and we aim to focus on those, specifically the change in the ocean's interactome with depth.

Results from co-occurrence analyses are highly dependent on the methods, but this is also the case with analyses of amplicon sequencing (ASVs, ESVs, zOTUs, OTUs, Swarms, etc), based on different regions of the rRNA gene, and using PCR or not (mTags). But, we consider that, despite these differences, the main patterns should be captured by the different methodologies.

Action: We have elaborated on interpreting the ecological network indices from an ecological perspective and the potential ecological implications of interactome change with depth. The modified paragraph is included below.

Line 530 “Network topologies changed with depth. Deep-sea networks were more clustered (higher transitivity) and had higher average path lengths (average number of steps (or "edges") that must be traversed to go from one node to any other node in the network), displayed stronger associations, and lower degree assortativity (nodes are less likely to associate with other nodes with similar degree) than surface networks. These topological changes may have multiple ecological implications. An increased clustering suggests more specialized or tightly-knit ecological relationships in the deep ocean, pointing to niche-specific microbes and interactions that could be potentially vulnerable to ocean change. For example, if a key species were to be lost, it could have a cascading effect on the entire community³⁸. This aligns with a hypothesis indicating a high niche partitioning and more specialist taxa in the deep ocean^{59,60}. A higher average path length indicates a more complex or fragmented network structure that could affect, for example, the transfer of carbon and energy through the community⁶⁵. Stronger positive associations among deep-sea microbes imply more stable and persistent interactions, likely influenced by the stable environmental conditions in the deep ocean⁴⁷. Yet, these interaction types could make communities more vulnerable to environmental changes via cascade effects³⁸. Lastly, the lower assortativity in deep-sea networks implies a more heterogeneous structure, with

microbes potentially interacting across a broader range of functional roles, metabolisms, or ecological niches. Overall, microbial networks in the deep sea seem more complex than surface counterparts in the tropical and subtropical ocean. A recent study investigating microbial networks in the western Pacific Ocean from 0 to 2000 m also found clear differences between aphotic and photic networks³¹. Furthermore, in agreement with our results, the previous study found that deep-sea networks had fewer edges and a higher average path length, pointing to looser connectivity than surface networks. Yet, contrary to our results, the study³¹ reports a higher clustering (transitivity) of photic networks compared to aphotic. This discrepancy could be linked to different sampling and network construction strategies, differences in the delineation of depth layers, total sampling depth, or the specific region that was investigated. In any case, our results and those from the mentioned work indicate a clear change in the ocean interactome as we dive into the deep ocean, which aligns with the observed vertical community changes^{12,15,37}, and which could imply different effects in the functioning of surface and deep-sea microbial ecosystems⁶⁵.”

I. 358-361: Could you find more “endemic” interactions when you run co-occurrence network inference for each region/depth-layer individually? As you stated before, interactions can change, but this would be less detectable with a global network. You are welcome to discuss that point further.

Answer: It is possible that running co-occurrence network analyses independently for each region or depth layer could increase the number of “endemic” interactions. The number of endemic interactions could also change when changing the parameters we have used during network construction or using different tools. We could though be confident that the endemic interactions captured by our global network are most likely robust, as they derive from the entire dataset.

Action: Now, we mention in the text:

Line 604 “How many of the regional associations that we detected represent endemic interactions needs further investigation. Even though our dataset is one of the largest generated so far for the tropical and subtropical global ocean and the Mediterranean Sea, it includes different sampling efforts for regions and depth layers. This may have introduced biases in the detected regional or global associations. Furthermore, different methodologies should be tested, as network construction tools, analytical thresholds, sequencing depth, and sampling design could all influence the amount of putatively endemic interactions that are detected. Despite these limitations, we consider that our work provides important insights into the amount of regional and cosmopolitan putative interactions in the tropical and subtropical global ocean and the Mediterranean Sea.”

I. 366-372: The conclusions are weak. What do your data really tell as new insights? What do you mean with “specific biogeographies which do not mirror taxonomic biogeographies”? What are you referring to here?

Answer & Action: We have modified the conclusions to indicate more explicitly the new insights derived from this work. The following sentence, “specific biogeographies which do not mirror taxonomic biogeographies” has been removed. The modified conclusion now reads (**Line 613**):

“To conclude, we have analyzed the spatial distribution of potential microbial interactions in the tropical and subtropical global ocean and the Mediterranean Sea, considering archaea, bacteria, and picoeukaryotes from surface to bottom waters. Thus, our work significantly expands previous efforts that focused on analyzing the microbial interactome of upper water layers of the global ocean^{27,29}. We have used an innovative approach, sample-specific networks, that allowed us to analyze the change of networks across locations and determine global and regional microbial associations across water layers. Therefore our work contributes to understanding the dynamics of the ocean interactome, still a developing research field⁷³. We found few global associations for the horizontal dimension of the ocean. In turn, 11-36% of the associations were regional and limited to specific depth layers. For the vertical dimension, our results indicate that associations change across the water column and that they may have specific depth distributions. Previous studies have investigated the horizontal and vertical turnover of the ocean microbiome (that is, community composition)^{5,6,8,12,15}. Our results expand this knowledge by indicating how potential interactions may change over the vertical and horizontal dimensions of the ocean. Furthermore, our results contribute to understanding the biogeography of potential interactions and provide hints on the links between different network topologies and ecosystem functioning, which is relevant in the context of global change⁷⁴. Finally, the identified widespread microbial associations, which could be important for ocean ecosystem function across multiple locations, could be the initial target of future monitoring or conservation efforts to preserve the ocean microbiome and interactome⁷⁵.”

I. 394: must be assigned taxonomically.

Answer: ASVs were assigned taxonomy using different databases. It is unclear to us if an action is requested.

I. 398-400: How do you account for a possible bias due to different size-fractions in network inference? Different size-fractions may have different (relative) abundances for the same taxa due to exclusion or accumulation by filter clogging of certain groups of organisms.

Answer: Most of our dataset considered samples from the 0.2-3 μm size-fraction. The *MalaDeep* dataset (13 samples) contained two size fractions, the 0.2-0.8 μm and the 0.8-20 μm . We think the impact of filter clogging is minimal in our dataset, as the small size-fraction (0.2-0.8 μm) was used only for prokaryotes, while the larger one (0.8-20 μm) was used only for microbial eukaryotes. Microbial eukaryotic signal from free-living cells is not expected in the small fraction (0.2-0.8 μm). Therefore, if eukaryotic signal was present in that fraction, it was not used (as this could correspond to broken cells). Similarly, the signal of prokaryotes in the larger size fraction (0.8-20 μm) is not used, as this could correspond to particle-attached cells, which we are not considering, or cells that ended up in that fraction due to filter clogging.

The above is indicated in the main text:

Line 676: “Given the cell sizes of prokaryotes versus microeukaryotes, we used the smallest size-fraction (0.2-0.8 μm) for prokaryotes and the larger one (0.8-20 μm) for microbial eukaryotes.

I. 412-413: How many ASVs were included in the analysis (after filtering)?
Mention here the number of ASVs that were used as input for flashweave

Answer:

The number of ASVs included in the analyses after filtering were:

- For eukaryotes: 2922 ASVs, representing 79.8% of the original reads.
- For prokaryotes: 2535 ASVs, representing 84.8% of the original reads.

In total, 5457 ASVs were used as input for Flashweave

This information is also provided at Lines 200-215 in the following script that is part of the companion GitHub:

https://github.com/InaMariaDeutschmann/GlobalNetworkMalaspinaHotmix/blob/main/00_Tables.Rmd

Action:

We have added the following information to the main text:

Line 693: “This filtering step left 2922 eukaryotic ASVs, representing 79.8% of the original eukaryotic reads, and 2535 prokaryotic ASVs, representing 84.8% of the original prokaryotic reads. ”

Line 697: “Second, we inferred a (preliminary) network using FlashWeave ⁴⁹, based on 5457 ASVs selecting the options “heterogeneous” and “sensitive”.

L. 424: Please elaborate on how EnDED works (how it identifies indirect edges). The number of indirect edges seems very low, especially that of temperature.

Answer:

One of the reasons for the low number of detected indirect edges is that FlashWeave already detects and removes a good fraction of them. To remove potentially remaining indirect edges, we applied EnDED on top of FlashWeave results. In other analyses, where EnDED was tested, the number of environmentally driven edges linked to temperature was 1920 (6.44%), with 37.8% being positive and 62.2% being negative edges, Supplementary Table 4 in ¹³.

In the Ph.D. thesis of Ina M. Deutschmann ¹⁸, some further analyses have been presented:

- Part II “Disentangling marine microbial association networks”; Chapter 8 “Further investigations including EnDED”; section “Comparing the application of EnDED on networks constructed with different tools using the Malaspina Surface data”
 - Applying EnDED to networks constructed with different tools indicated that all tools were prone to indirect dependencies, at least to a minor fraction.
 - We found fewer indirect dependencies among associations inferred by three tools (SparCC, MICtools, and FlashWeave) than associations inferred by one or two methods.
- Part III “Further discussion and thesis conclusions”; Chapter 9 “Environmentally-driven associations”; Technical aspects of environmentally-driven associations
 - The effect of environmentally-driven edges is prominent in association networks. All tested network construction tools were prone to the effects of indirect dependencies, but the extent of specific environmental drivers differed among tools. Thus, comparing environmental-drivers in microbial ecosystems will require the same sampling, environmental data, filter, and network construction strategies.

Action:

We have added to the main text a brief explanation of EnDED, and the reader is referred to the main publication ¹³ of this approach for additional details.

Line 704: “Thus, we applied EnDED (Environmentally Driven Edge Detection) ⁹¹, a method suitable for large-scale spatial data designed to identify which links between microorganisms in an association network are environmentally driven. The program implements four methods: Sign Pattern, Overlap, Interaction Information, and Data Processing Inequality ⁹¹. EnDED can use these methods individually or in combination to better predict environmentally driven associations in microbial networks.”

I. 466-467: What is the difference between point i) and ii) ? Also, please elaborate on the difference between Supplementary Figure 5A and B. Please also elaborate how many ASVs/Edges were filtered out using that procedure.

Answers:

The difference between points i and ii is the following:

1. *“i) how many edges were kept in the random subsamples compared to all samples”*

-“kept edges in random subsample”: here, we took 90%, 80%, ... 10% of the samples randomly (1000 times). So let’s say we took 50% of 30 samples, so 15 samples. Then, we apply the Jaccard filtering of 20%. And 200 edges remain.

- "kept edges with all samples": This is the number of samples remaining if a Jaccard index of 20% is applied (as done for primary analysis) on all samples. So, for example, in 30 samples. Let's say here 400 edges remained.

- Then, the fraction would be $200/400 = 0.5$

2. *"ii) how many edges were kept in the random subset that were also kept when all samples were used."*

- "kept edges in random subsample": here, we took 90%, 80%, ... 10% of the samples randomly (1000 times). So let's say we took 50% of 30 samples, so 15 samples. Then, we apply the Jaccard filtering of 20%. And 200 edges remain (set A)

- "kept edges with all samples": this is the number of samples remaining if a Jaccard index of 20% is applied (as done for primary analysis) on all samples. So, for example, in 30 samples. Let's say here 400 edges remained (set B)

- Now, we look at the intersection of A and B and not solely the number of edges in A

Supp. Figures 5A and 5B reflect the two points above as well as the proportion of edges kept.

Action:

We clarified the previous points in the main text:

Line 754: "Lastly, requiring $J > 20\%$, we evaluated the robustness of the third condition (i.e., the association has a region and depth-specific Jaccard index, J_{RL} , above 20%), for generating sample-specific subnetworks for each region and depth with sufficient samples. Within each region and depth, the samples were randomly subsampled, containing 10% to 90% of the original set using all samples. We determined the fraction of edges kept in the subsampled set compared to the original set. Specifically, we determined i) how many edges were kept in the random subsamples compared to all samples (that is, only the number of kept edges) and ii) how many edges were kept in the random subset that were also kept when all samples were used (that is, which edges were kept). We repeated the procedure for each region-depth combination 1000 times."

Reviewer #2 (Remarks to the Author):

Dear Authors and Editor,

Thank you for the opportunity to review the manuscript “Disentangling microbial networks across pelagic zones in the global ocean” by Deutchmann and colleagues, submitted to Nature Communications.

This manuscript makes the novel and important observation that the patterns of statistical associations between microorganisms generally differ between ocean regions and depths. In particular, they note that there are few globally consistent statistical associations, and few that are present across all depths – rather most associations are geographically local.

I think that this is an important point, and by itself makes this manuscript worthy for publication in Nature Communications. Additionally this is, to my knowledge, the first comprehensive network analysis approach applied to Malaspina data (combined with the Hotmix expedition) which is also an important step.

Answer: We thank the reviewer for their time and the comments that helped to improve our work.

However I had some challenges understanding much of their methods, as well as the the significance of some of their figures and points. I think the manuscript could be made more readable with consideration of my flowing comments:

Line 33: “that do not correspond to taxonomic distributions” What does it mean that heterogenous distributions do not respond to taxonomic distributions. I presume “distributions” mean something different in each case. In the former, its that interactions differ by ocean basin and depth. I’m not sure what “distributions” mean in the later case.

Answer & Action: We have removed this sentence as it generated confusion and is not crucial to conveying the main message.

Line 38: What does “Despite microbial sinking” mean in this context. Are they detecting microbial sinking? Or do they contend that one would expect that since microbes are transported by marine snow, one would see the same interactions throughout the water column. If the later, I don’t see why microbial transport would be expected to lead to different inteeractions. Could “despite microbial sinking” be removed? Is so I think the sentence would be just as strong.

Answer & Action: Thank you for this comment. This has been changed to (Line 38) “despite microbial vertical dispersal”

Different studies have shown vertical connectivity in the ocean microbiome, which is in part presumably linked to sinking particles ^{8,19}. Sinking particles, their associated microbes, and their

interactions could be transported from the surface to the ocean interior, but what we observe is that “most surface water associations do not persist in deeper ocean layers”

Line 39: I kind of like calling this the interactome. Great buzzword, makes network analysis seem like its on the -omics train. 100% in favor. However, I'm a little nervous that the authors aren't really detecting interactions. They are actually detecting statistical correlations. Which might be related to interactions. Maybe some justification of this term, somewhere in the body of the paper is in order. Then I the rest of us networks people can start referring to our own networks as “interactome” studies, cite this one and get in on that -omics credibility.

In any case, they do a pretty good job of this on line 62, but should make the case for calling it an interactome there too. Or they could call these something other than an interactome (which I am not requesting).

Answer & Action: We thank the reviewer for this comment. We have been especially careful in indicating that we are working with associations representing potential interactions. We have checked again the entire text to prevent confusion. In addition, to clarify this point, We have added the following sentence (**Line 78**): "Here, we refer to this set of potential ecological interactions based on association networks as the “interactome”, similarly to other works ^{27,29}."

Line 69: Somebody at some point should do an analysis like this one including the TARA oceans data in addition to this data set. Somewhere the authors should explain why they excluded this publicly available information.

Answer & Action: The main reason for not including datasets from TARA Oceans is the heterogeneity of the data. We have based our analyses on amplicons from specific regions of the 16S (V4-V5) and 18S (V4) rRNA gene maker, and we spent a substantial amount of work to ensure that the datasets we have included are compatible. Publicly available rRNA gene datasets from TARA oceans are based on miTags (the rRNA gene is extracted from metagenomes) or other rRNA regions (V9 region). Even though it would be feasible to mix different markers in network analyses, we preferred to base our analyses on the same markers to prevent possible biases.

We have added the following sentence to the Methods section, **Line 657**: “Publicly available datasets from the global campaign TARA Oceans^{27,29} were not considered due to differences in the used methodologies to obtain the data (e.g., miTags vs. amplicons⁴⁸ and different marker regions of the rRNA gene).”

Somewhere around line 74: the authors should cite Kara et al. 2012. This team broke spatial analysis of one location up by season and showed that the ... interactome differed between times of the year. Their own reference 33 does this as well, of course.

Kara EL, Hanson PC, Hu YH, Winslow L, McMahon KD. A decade of seasonal dynamics and co-occurrences within freshwater bacterioplankton communities from eutrophic Lake Mendota, WI, USA. ISME J. 2012 Oct 11;7:680–4.

Answer & Action: We thank the reviewer for this suggestion, we have incorporated the reference in the new version of the manuscript.

Line 98: “Recent approaches we developed...” A short overview of how these approaches work would improve readability and not require all readers to find the other manuscript.

Action: We have added the following text to the introduction:

Line 117 “In a nutshell, each subnetwork (sample) includes only nodes and edges present in the overarching static network. Three key conditions must be met for an edge to be included in a subnetwork: 1) the edge must already exist in the single static network, 2) both microorganisms connected by the edge must have a sequence abundance above zero in the sample, and 3) the microorganisms must appear together in more than 20% of the samples for a specific marine region and depth. ”

Line 114: Unless its a requirement of the journal, I think the paper might read better in the usual order with methods before results. The methods are novel, and kind of necessary to understand the results, I don't think the order is critical though, so the authors are welcome to disregard this suggestion.

Answer: The format is a requirement of the journal

Line 238: Misplaced “)”

Answer: Thank you. Still, the position of the “)” is as intended.

Line 246: What sorts of future designs could detect weak interactions?

Action: We have added the following sentence to the main text:

Line 331 “For example, high-frequency sampling campaigns including multiple replicates, microcosms and manipulative field experiments, and metabolic modeling coupled to metabolomics⁴⁵.”

Line 274: Has it been established that the deep ocean system has “higher complexity”. If so, in what way?

Answer & Action: We have tried to clarify this section, it now reads:

Line 411 “Even though environmental gradients in the deep ocean are smooth when compared to the surface⁴⁸, specific processes could contribute to increasing the complexity and number of niches in the deep ocean, potentially leading to an increase in the number of regional associations. For example, different niches may be associated with the quality and types (labile vs. recalcitrant) of organic matter reaching the deep ocean from the epipelagic zone³⁰, which is significantly different across oceanic regions⁵¹. In an exploration of generalists versus specialist prokaryotic metagenome-assembled genomes (MAGs) in the Arctic Ocean, most specialists were linked to mesopelagic samples, indicating that their distribution was uneven across depth layers⁵².”

Around Line 350: I encourage the authors to discuss how the patterns in the Mediterranean Sea might be representative of patterns in seas generally, as opposed to specific to the Mediterranean Sea. I realize they have not surveyed other seas and this is an oceans + med sea paper.

Answer & Action: Given that we do not have data from other seas, we can not discuss much on how the patterns we found differ from other seas. Yet, we have added the following sentence to the discussion:

Line 595 “Our findings from the Mediterranean Sea suggest that other enclosed or semi-enclosed seas, such as the Baltic Sea, Black Sea, Caspian Sea, and Red Sea, could harbor endemic or regional microbial interactions.”

Line 389: Don't 515F-Y and 926R also get eukaryotes? How did they handle the overlapping eukaryotic data.

Answer: Eukaryotic data has been removed through fragment removal and *in silico*. Only the data derived from the primers TAREukFWD1 and TAREukREV3 has been used to analyze the eukaryotes. One reason not to use the eukaryotic data from the primers 515F-Y and 926R is that forward and reverse reads usually do not overlap.

391. “in 5” is not grammatical.

Action: corrected

Line 444. “Third, microorganisms needed to appear together (intersection) in more than 20% of the samples.” Why? This seems like it would select against strong negative associations. One could imagine if two species just can't co-occur with each other for some reason, then this filter would make them not associated. Thus all of the negative associations must be weak and have exceptions to make it through the analysis. Perhaps this is why they have so many positive edges.

Answer:

Thank you for your comment. The 20% threshold was implemented to enhance the signal-to-noise ratio in our heterogeneous dataset. Our aim was to focus on robust associations. We have tested different thresholds below and above the chosen 20% (the amount of remaining edges with each filtering threshold is listed in Supplementary Table 5). We consider that choosing smaller thresholds will not change the results substantially, as the differences are not large. Nevertheless, it is possible that some strong negative associations have been missed, but in any case, those would have been more difficult to support, as such strong co-exclusions could be reflecting other processes (e.g., different selective pressures, associations with other organisms, etc.). Exhaustive sensitivity analyses on the recovery of strong negative associations on different thresholding criteria are very interesting but beyond the scope of this work.

Line 501: "as proposed in 83. " is not grammatical as written. Add the Yaveroğlu et al. before the superscripted number.

Action: corrected

Figure 2: How is the layout of the main network defined? How did the authors arrange the nodes so that the edges tend to cluster by region, or does that happen automatically? What is "association prevalence" as written, I don't understand the x-axis of the histograms.

Answer: The layout of the main network is indicated in Methods: "single static network, visualized in Gephi ⁹⁰ v.0.9.2, using the Fruchterman Reingold Layout ⁹¹ with a low gravity score of 0.5." In short, we used the Fruchterman Reingold Layout to visualize the single static network. Then, we used this network (so to say, "fixed /froze" the layout) and only changed the edges and the color. All nodes are kept in the network to keep the network layout the same among regions and depths.

The arrangement is automatic via the Fruchterman Reingold Layout. However, each time the algorithm is applied, a slightly different visualization will appear. Nevertheless, overall patterns remain, and highly connected areas tend to be visualized closer to each other

Regarding prevalence, this is defined in the Methods section "To determine an association's spatial recurrence, we calculated its prevalence as the fraction of subnetworks in which the association was present. We determined association prevalence across the 397 samples and each region-layer combination. We mapped the scores onto the single static network"

Action: We have clarified the definition of prevalence in the legend of Figure 2, **Line 1085** "A) Association prevalence, that is, the fraction of subnetworks (samples) in which an association appeared considering all depth layers across the tropical and subtropical global ocean and the Mediterranean Sea."

Figure 3: I don't entirely follow what is happening in this figure. What are the nodes and edges exactly? They aren't described in the legend.

Answer: Thanks for pointing this out. The nodes (outer chord) are microbial taxa (ASVs) grouped by taxonomic rank. The edges (connections inside the chord circle) represent the associations between the ASVs.

Action: We have added the following information to the figure legend.

Line 1101 “The nodes (outer chord) are microbial taxa (ASVs) grouped by taxonomic rank. The edges (connections inside the chord circle) represent the associations between the ASVs. Note how the proportion of associations changes across the vertical and horizontal dimensions of the ocean.”

Figure 5: Requires clean up. The facet labels and text are all very small. There are many edges but they are impossible to distinguish from each other. Frankly, I’m not sure what I’m supposed to be taking away from this figure. Indeed, the figure caption, as in Figure 3, doesn’t really explain what is going on, and the reader is left to guess.

Answer: Thanks for indicating this.

Action: We have now modified this figure following the suggestions. We have also added the following text to the figure legend (**Line 1128**): “Note that few associations were present throughout the water column within a region and that most associations from the meso- and bathypelagic did not appear in the upper layers except for the MS and NAO.”

Figure 6: Warrants more explanation. I could really use a plain language summary, or even a somewhat technical one of what a minimal spanning tree is, and why it is informative. I like the idea of networks about networks and they look neat, but I don’t really follow them.

Answer & Action: We have substantially expanded the legend of Figure 6, explaining the Minimum Spanning Tree and its interpretation, as indicated below:

Line 1132 “**Figure 6: Similarity of sub-networks based on topology.** The figure shows the Minimal Spanning Tree (MST) based on dissimilarities in sub-network topologies (see Methods). Each of the 397 nodes in the MST represents a unique subnetwork derived from specific samples. The 396 edges in the MST represent the dissimilarity between sub-network pairs (samples). The edge weights are assigned based on the network-dissimilarity score. A Minimal Spanning Tree aims to connect all 397 nodes to one connected tree under the condition that the 396 edges have the lowest sum of scores. Note there may be more than one solution for an MST. The MST allows simplifying the main similarity patterns between subnetworks, i.e., nodes closer to each other indicate that sub-networks are more similar in topology and vice-versa. Thus, the shown MST allows a broad comparison of sub-networks based on their similarities in interconnection patterns or topologies. The similarity in topology suggests that similar ecological processes, environmental conditions, geographic features, or a mix of them have influenced the microbial communities and

their potential interactions in these subnetworks. Thus, we have mapped on top of the MST geographic, oceanographic, and environmental variables. The rationale is that if nodes that cluster close to each other in the MST also share similar magnitudes in some of these variables, then such variables could be influencing the topology of the sub-networks (for example, in panel A, multiple sub-networks from the surface ocean are grouping, indicating that they tend to be more similar among themselves than to sub-networks from other depth layers). Nodes are colored according to A) the sample's depth layer, B) the sample's ocean region, C) the subnetworks cluster (see Supplementary Table 2), and D) selected environmental variables. In C), the bar plots indicate the different layers, colored as in A), within each cluster. For each cluster, we show how many subnetworks belong to Surface, DCM, Mesopelagic, and Bathypelagic. The cluster is indicated by color (below bars and on top of the MST) and cluster number (x-axis). The depth of samples (Surface, DCM, Mesopelagic, and Bathypelagic) within clusters is indicated by the colors in the bars (see color code in panel A) and the number of different depth layers considered in each cluster by the bars' height (y-axis). ”

Reference 33 is published. Please cite main paper.

Action: The reference has been updated.

REFERENCES

1. Chaffron, S. *et al.* Environmental vulnerability of the global ocean epipelagic plankton community interactome. *Sci Adv* **7**, (2021).
2. Lima-Mendez, G. *et al.* Determinants of community structure in the global plankton interactome. *Science* **348**, 1262073 (2015).
3. Salazar, G. *et al.* Gene Expression Changes and Community Turnover Differentially Shape the Global Ocean Metatranscriptome. *Cell* **179**, 1068–1083.e21 (2019).
4. Logares, R. *et al.* Disentangling the mechanisms shaping the surface ocean microbiota. *Microbiome* **8**, (2020).
5. De Vargas, C. *et al.* Eukaryotic plankton diversity in the sunlit ocean. *Science* **348**, (2015).
6. Villarino, E. *et al.* Global beta diversity patterns of microbial communities in the surface and deep ocean. *Glob. Ecol. Biogeogr.* **31**, 2323–2336 (2022).
7. Giner, C. R. *et al.* Marked changes in diversity and relative activity of picoeukaryotes with

- depth in the world ocean. *ISME J.* **14**, 437–449 (2020).
8. Mestre, M. *et al.* Sinking particles promote vertical connectivity in the ocean microbiome. *Proc. Natl. Acad. Sci. U. S. A.* **115**, E6799–E6807 (2018).
 9. Krabberød, A. K. *et al.* Long-term patterns of an interconnected core marine microbiota. *Environ Microbiome* **17**, 22 (2022).
 10. Deutschmann, I. M. *et al.* Disentangling temporal associations in marine microbial networks. *Microbiome* **11**, 83 (2023).
 11. Röttjers, L. & Faust, K. From hairballs to hypotheses – biological insights from microbial networks. *FEMS Microbiol. Rev.* 1–20 (2018).
 12. Latorre, F. *et al.* Niche adaptation promoted the evolutionary diversification of tiny ocean predators. *Proc. Natl. Acad. Sci. U. S. A.* **118**, (2021).
 13. Deutschmann, I. M. *et al.* Disentangling environmental effects in microbial association networks. *Microbiome* **9**, (2021).
 14. Schmidt, T. S. B., Matias Rodrigues, J. F. & von Mering, C. A family of interaction-adjusted indices of community similarity. *ISME J.* **11**, 791–807 (2017).
 15. Chow, C. E., Kim, D. Y., Sachdeva, R., Caron, D. A. & Fuhrman, J. A. Top-down controls on bacterial community structure: microbial network analysis of bacteria, T4-like viruses and protists. *ISME J.* (2014) doi:10.1038/ismej.2013.199.
 16. Coyte, K. Z., Schluter, J. & Foster, K. R. The ecology of the microbiome: Networks, competition, and stability. *Science* **350**, 663–666 (2015).
 17. Tackmann, J., Matias Rodrigues, J. F. & von Mering, C. Rapid Inference of Direct Interactions in Large-Scale Ecological Networks from Heterogeneous Microbial Sequencing Data. *Cell Systems* **9**, 286–296.e8 (2019).
 18. Deutschmann, I. M. Disentangling ecological networks in marine microbes. (Universitat Politècnica de Catalunya (UPC), 2021).
 19. Ruiz-González, C. *et al.* Major imprint of surface plankton on deep ocean prokaryotic

structure and activity. *Mol. Ecol.* **29**, 1820–1838 (2020).

REVIEWERS' COMMENTS

Reviewer #1 (Remarks to the Author):

The authors have thoroughly revised the manuscript and considered my recommendations, suggestions and critique. Hence the revised version greatly improved and reads much clearer and more self-critical. There are two points left which need to be considered.

Most recently, presumably after resubmission of this revision, a very interesting publication came out in Nature Communications which took the opposite approach to this study, not looking at station- or region-specific co-occurrence/association networks of pelagic prokaryotes but looking at globally occurring networks in the global oceans, Milke et al., Nature Comm 14: 6141. These authors, applying their approach to global data sets from tropical to subpolar latitudes and including Malaspina data, show that prokaryotic communities in pelagic oceans are composed of co-occurring modules and that surface currents are important in community assembly. It is timely and most appropriate that this resubmitted study compares and contrasts both approaches and outcomes. It may shed more light on the globally occurring network associations found in this study which greatly dominate the network associations.

As another minor point: Taxonomic names from l. 165 to 175 need to be italicised.

Reviewer #2 (Remarks to the Author):

The authors have adequately responded to my comments. I thank them for their thorough revisions and producing this excellent manuscript.

Reviewer #3 (Remarks to the Author):

We thank both reviewers again for their time and the comments and suggestions to improve our work. Please find below our point-by-point answer to the reviewer's comments (in blue) and the actions taken, including changes in the main text. We also provide a version of the manuscript with tracked changes as a related manuscript file. Please note that the indicated lines of the changes refer to the document with tracked changes.

REVIEWERS' COMMENTS

Reviewer #1 (Remarks to the Author):

The authors have thoroughly revised the manuscript and considered my recommendations, suggestions and critique. Hence the revised version greatly improved and reads much clearer and more self-critical.

We thank the reviewer for their comments.

There are two points left which need to be considered.

Most recently, presumably after resubmission of this revision, a very interesting publication came out in Nature Communications which took the opposite approach to this study, not looking at station- or region-specific co-occurrence/association networks of pelagic prokaryotes but looking at globally occurring networks in the global oceans, Milke et al., Nature Comm 14: 6141. These authors, applying their approach to global data sets from tropical to subpolar latitudes and including Malaspina data, show that prokaryotic communities in pelagic oceans are composed of co-occurring modules and that surface currents are important in community assembly. It is timely and most appropriate that this resubmitted study compares and contrasts both approaches and outcomes. It may shed more light on the globally occurring network associations found in this study which greatly dominate the network associations.

We thank the reviewer for pointing us to this interesting paper that was published during the review of our manuscript. We have now added the following paragraph to the Discussion, comparing both approaches:

L376: "Recently, Milke and colleagues⁵² have reported that prokaryotic communities in the epipelagic Pacific, Atlantic, and southern Indian Oceans and the Mediterranean Sea are structured into modules of co-occurring taxa with specific distributions and environmental preferences. In contrast, our work extends beyond the epipelagic zone to include deeper layers and microbial eukaryotes, emphasizing the dynamic nature of potential interactions across the horizontal and vertical dimensions of the tropical and subtropical global ocean and the Mediterranean Sea. Thus, while Milke *et al.*'s work underscores the importance of specific co-occurring taxa across vast oceanic regions, our analyses focus on the change of potential interactions with depth and ocean basin. Both works complement each other by indicating

that marine microbial communities include both stable, tightly-knit associations as well as associations that are more idiosyncratic or that can change over space and time³⁶.”

We added to the bibliography:

“Milke, F., Meyerjürgens, J. & Simon, M. Ecological mechanisms and current systems shape the modular structure of the global oceans’ prokaryotic seascape. Nat Commun 14, 6141 (2023)”

As another minor point: Taxonomic names from l. 165 to 175 need to be italicised.

Thanks for this comment. We have not italicised these taxonomic names as the journal requires to italicise species names only.

Reviewer #2 (Remarks to the Author):

The authors have adequately responded to my comments. I thank them for their thorough revisions and producing this excellent manuscript.

We thank the reviewer for their comments.

Reviewer #3 (Remarks to the Author):
